# Dual functions of SPOP and ERG dictate androgen therapy responses in prostate cancer

Tiziano Bernasocchi[1,2,11], Geniver El Tekle [1,2,11], Marco Bolis[1,11], Azzurra Mutti[1], Arianna Vallerga[1], Laura P. Brandt[3], Filippo Spriano[1,2], Tanya Svinkina[4], Marita Zoma[1,2], Valentina Ceserani[1], Anna Rinaldi[1], Hana Janouskova[1], Daniela Bossi[1], Manuela Cavalli[1], Simone Mosole[1], Roger Geiger [5], Ze Dong[6], Cai-Guang Yang [6], Domenico Albino[1], Andrea Rinaldi[1], Peter Schraml[7], Simon Linder[8], Giuseppina M. Carbone [1], Andrea Alimonti [1], Francesco Bertoni[1], Holger Moch[7], Steven A. Carr[4], Wilbert Zwart [8], Marianna Kruithof-de Julio [9,10], Mark A. Rubin [3], Namrata D. Udeshi [4] & Jean-Philippe P. Theurillat [1✉]

Driver genes with a mutually exclusive mutation pattern across tumor genomes are thought to have overlapping roles in tumorigenesis. In contrast, we show here that mutually exclusive prostate cancer driver alterations involving the *ERG* transcription factor and the ubiquitin ligase adaptor *SPOP* are synthetic sick. At the molecular level, the incompatible cancer pathways are driven by opposing functions in SPOP. ERG upregulates wild type SPOP to dampen androgen receptor (AR) signaling and sustain ERG activity through degradation of the bromodomain histone reader ZMYND11. Conversely, SPOP-mutant tumors stabilize ZMYND11 to repress ERG-function and enable oncogenic androgen receptor signaling. This dichotomy regulates the response to therapeutic interventions in the AR pathway. While mutant SPOP renders tumor cells susceptible to androgen deprivation therapies, ERG promotes sensitivity to high-dose androgen therapy and pharmacological inhibition of wild type SPOP. More generally, these results define a distinct class of antagonistic cancer drivers and a blueprint toward their therapeutic exploitation.

[1] Institute of Oncology Research, Faculty of Biomedical Sciences, Università della Svizzera italiana, 6500 Bellinzona, TI, Switzerland. [2] University of Lausanne, 1011 Lausanne, VD, Switzerland. [3] Department of Biomedical Research, University of Bern, 3008 Bern, Switzerland. [4] The Broad Institute of MIT & Harvard, Cambridge, MA 02142, USA. [5] Institute for Research in Biomedicine, 6500 Bellinzona, TI, Switzerland. [6] State Key Laboratory of Drug Research, Shanghai Institute of Materia Medica, Chinese Academy of Sciences, 201203 Shanghai, China. [7] Department of Pathology and Molecular Pathology, University Hospital Zurich, 8091 Zurich, ZH, Switzerland. [8] The Netherlands Cancer Institute, Oncode Institute, 1066 CX Amsterdam, The Netherlands. [9] Department of Biomedical Research, Urology Research Laboratory, University of Bern, 3010 Bern, Switzerland. [10] Department of Urology, Inselspital, Bern, Switzerland. [11] These authors contributed equally: Tiziano Bernasocchi, Geniver El Tekle, Marco Bolis. ✉email: Jean-Philippe.Theurillat@ior.usi.ch

Normal cells transform into cancer cells by the acquisition of genetic aberrations in so called driver genes. In some instances, the functional redundancy of mutations in different genes results in a mutually exclusive mutation pattern across tumor genomes because one alteration is sufficient to activate the specific oncogenic pathway. Based on this assumption, bioinformatic tools have been generated to search for functional redundancy of mutated genes in larger cancer genome datasets[1,2].

In prostate cancer, recurrent gene fusions involving the *ERG* transcription factor and point mutations in the ubiquitin ligase adaptor *SPOP* are two truncal mutations that are mutually exclusively distributed across tumor genomes (Fig. 1a and Supplementary Fig. 1a)[3–9]. The underlying cause for this exquisite pattern remains controversial. While earlier reports suggested a functional redundancy between mutant SPOP and ERG based on the finding that mutant SPOP stabilizes the ERG oncoprotein[10,11], more recent studies challenge this view by showing descriptive evidence for divergence in tumorigenesis[3,12].

Here, we show that *SPOP*- and *ERG*-mutant cancer subtypes are driven by antagonistic tumorigenic pathways involving fundamentally different roles of SPOP and androgen receptor signaling levels. The divergence in tumorigenesis is associated with specific susceptibilities to SPOP inhibition and perturbation in the androgen receptor pathway.

## Results

**Activation of the ERG oncogene and missense mutation in SPOP are synthetic sick.** To shed light on these recurrent driver genes' functional relationship, we assessed the impact of SPOP mutations and ERG activation on the cellular growth of mouse prostate epithelial organoids. To do so, we first established and validated the organoids by the presence of multilayered structure with the expression of CK5 and CK8 in basal and luminal cells, respectively (Supplementary Fig. 1b). In agreement with recent reports, lentiviral-transduced point mutants of SPOP (SPOP-Y87C, SPOP-W131G) or a truncated version of ERG, which typically results from gene fusion with androgen-regulated genes in prostate cancer (ΔERG, amino acids 33–486), promoted cell growth (Fig. 1b and Supplementary Fig. 1c)[13–16]. While SPOP mutant organoids displayed a round shape, ERG's over-expression gave rise to characteristic finger-like protrusions. Surprisingly, the joint expression of both drivers considerably diminished cell growth and reduced finger-like protrusions, implying a synthetic sick relationship between the two genetic alterations. Cytological follow-up analysis revealed reduced proliferation evidenced by reduced Ki-67 and increased p16, P-HP1γ positivity, and vacuolization of the cytoplasm compatible with senescence induction (Supplementary Fig. 1d).

We wondered if the observed synthetic sick relationship also applied to established cancer cells from advanced, castration-resistant metastatic disease. Forced expression of mutant SPOP (SPOP-Y87C, SPOP-W131G) promoted 3D growth of *ERG* fusion-negative LAPC-4 human prostate cancer cells (Supplementary Fig. 2a). The oncogenic effect was paralleled by an increase in the expression of the oncogenic transcription factors MYC and HOXB13 and a decrease in the cell cycle inhibitor p21 as seen in an organoid line derived from *Spop^F133V*-mutant transgenic mice (Supplementary Fig. 2a, b)[13]. In contrast, we observed the opposite phenotypic and molecular changes in VCaP human prostate cancer cells harboring the recurrent *TMPRSS2-ERG* fusion (Fig. 1c, d & Supplementary Fig. 2c, d). In this setting, mutant SPOP (SPOP-Y87C, -F102C, -W131G, -F133S) dramatically decreased cancer cells' proliferation in culture and the growth of xenograft tumor models in vivo.

Similarly, to the mouse prostate organoids, possible induction of senescence was evidenced by increased senescence-associated β-galactosidase (SA-β-gal)-positive cells and upregulation of p21 and GDF15 protein levels. In line with this, the transfer of conditioned medium from VCaP cells expressing mutant SPOP (SPOP-Y87C, SPOP-W131G) also reduced the proliferation of parental VCaP cells, indicating a possible contribution of senescence-associated secretory phenotype (SASP) and suggesting that senescence could be one of the possible biological pathways involved in the synthetic sick relationship (Fig. 1d and Supplementary Fig. 2f, j).

Conversely, forced expression of ΔERG significantly reduced the growth of SPOP-Y83C mutant LuCaP-147 patient-derived xenograft (PDX) cancer cells in vivo and in culture (Fig. 1e and Supplementary Fig. 3a, b)[17], adding orthogonal support for a synthetic sick relationship between mutant SPOP and ΔERG in advanced prostate cancer. Besides, the over-expression of MYC promoted cancer cell growth in both VCaP and LuCaP-147 cells (Supplementary Fig. 3c, d). The latter finding may suggest that the over-expression system per se is not the underlying cause of the synthetic sick relationship mentioned above.

Next, we wondered if genetic or pharmacologic suppression of ERG signaling may revert the growth suppressing function of mutant SPOP in VCaP cells. Indeed, knockdown of ERG by short-hairpin RNA interference decreased the growth of VCaP control cells and of cells over-expressing wild-type SPOP, while it promoted the growth of cells over-expressing SPOP-W131G (Supplementary Fig. 3e). In addition, low doses of the ETS inhibitor YK-4-279 promoted specifically the growth of VCaP cells overexpressing mutant SPOP (Fig. 1f). We noted a similar effect when VCaP cells were co-treated with a small molecule inhibitor of SPOP (Supplementary Fig. 3f)[18]. In aggregate, the data support an antagonistic relationship between oncogenic activation of ERG and a loss of SPOP function in prostate cancer cells.

**Mutant SPOP-induced androgen receptor signaling antagonizes ERG activity.** To assess the underlying molecular biology of the antagonistic relationship between *ERG*-fused and *SPOP*-mutants tumors, we interrogated the transcriptomes from the TCGA cohort to nominate differences across these tumor subtypes. Indeed, the unbiased principal component analysis (PCA) revealed significant differences in the transcriptional output (Fig. 2a). The differences were maintained in castration-resistant prostate cancers (CRPC) from the (SU2C) cohort using a single-sample gene-set enrichment analysis approach (Fig. 2b and Supplementary Data 1). Furthermore, derived (PDX) models also retained analog transcriptional differences, as demonstrated by different behavior shown by *SPOP*-Mutant (LuCaP-78, −147) and *ERG*-Fused (LuCaP-35, −23.1, VCaP cells) models (Fig. 2b and Supplementary Data 1).

In *SPOP*-mutant prostate cancer, several dysregulated SPOP substrates (e.g., NCOA3, TRIM24, BET proteins) have been shown to boost the AR pathway leading ultimately to high levels of AR target genes (Supplementary Fig. 4a)[3,13,19–26]. In contrast, *ERG*-fused cancer cells typically express lower levels of AR target genes as illustrated by the widely adopted AR score (Supplementary Fig. 4a)[3]. We further performed differential expression analysis between the two tumor subtypes to get more insights into this different behavior. Gene-set enrichment analysis resulted in a clear upregulation of the canonical androgen response pathway in *SPOP*-mutant vs. *ERG*-fused tumors, as defined by the respective Hallmark gene-set, curated by the Molecular Signature Database (MSigDB) (Supplementary Fig. 4b). In line with the divergence of transcriptome profile identified in the PCA plot of Fig. 2a, a clear

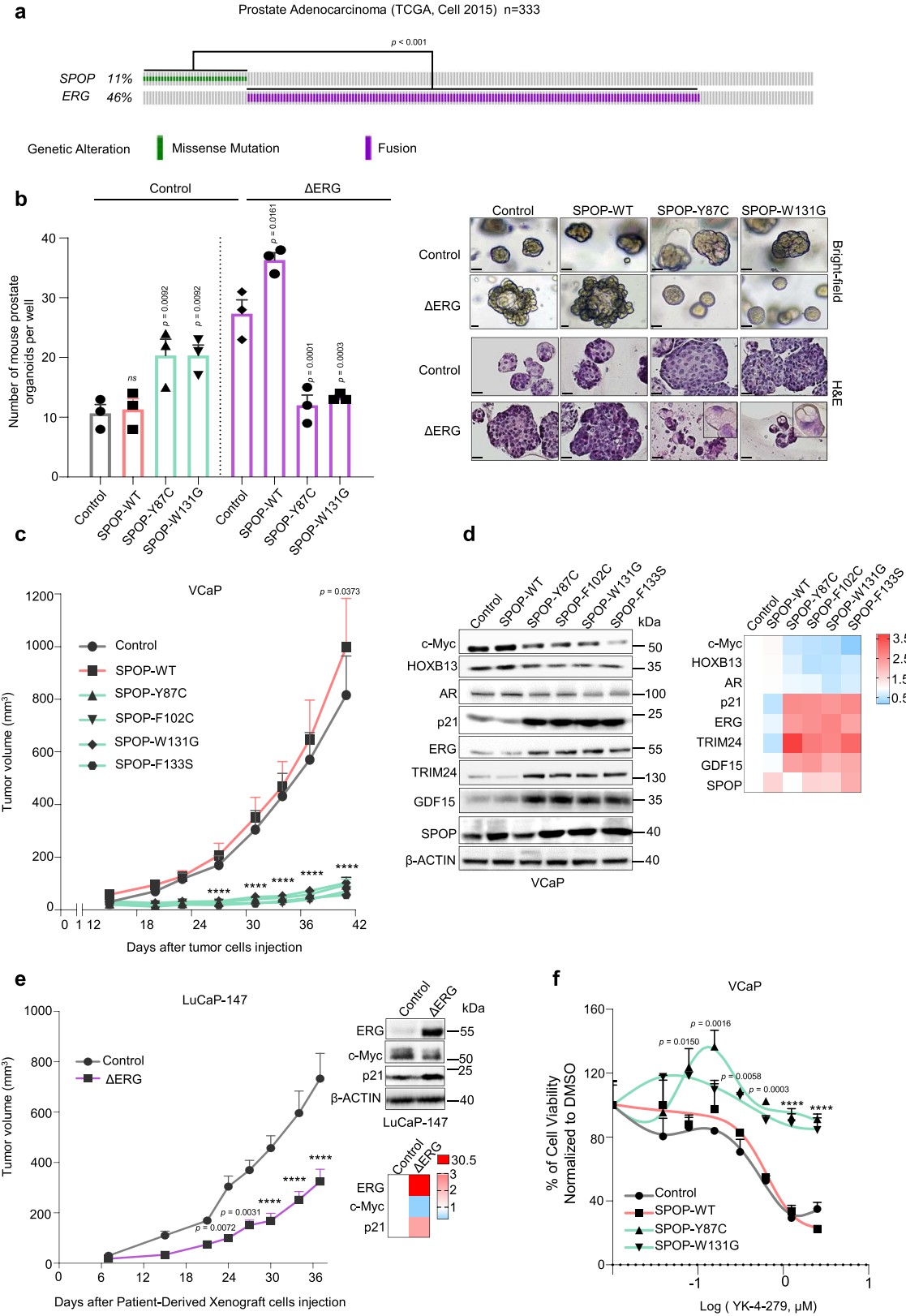

division between *SPOP*-mutant and *ERG*-fused tumors was also reported in their respective cistrome counterpart[25]. By re-analyzing ChIP-Seq data, we could determine that most differentially bound regions between both tumor types are characterized by increased AR-binding in *SPOP*-mutant tumors (Fig. 2c and Supplementary Data 2).

Next, we analyzed the transcriptome changes of the VCaP cells overexpressing SPOP mutants (SPOP-MTs; SPOP-Y87C, -F102C, -F133S). The unbiased hallmark analysis showed a dramatic increase in the androgen response, recapitulating the changes identified in primary prostate cancer (Supplementary Figs. 4b and 5a, b). Based on these results, we posited that differential levels of

**Fig. 1 Genetic alterations in *SPOP* and *ERG* are synthetic sick. a** Distribution of genetic alterations in *SPOP* and *ERG* across 333 primary prostate cancers in TCGA database[3,8,9]. **b** 3D growth of mouse prostate epithelial organoids derived from C57BL/6 mice over-expressing the indicated SPOP and ERG species (bar represents 20 μm) (n = 3, technical replicates). Representative bright-field pictures and hematoxylin and eosin (H&E) stained sections are shown. **c** In vivo growth of VCaP xenografts over-expressing the indicated SPOP species in immune-compromised mice (each group, n = 10). **d** Immunoblot of VCaP cells overexpressing the indicated SPOP species and corresponding quantification of the indicated protein levels depicted as a heatmap. Protein expression changes were normalized to β-ACTIN and Control cell line, (n = 2). **e** Tumor growth kinetics of xenografts established from LuCaP-147 PDX (SPOP-Y83C) stably overexpressing ΔERG or Control vector (each group, n = 10). Corresponding immunoblot and quantification are depicted as a heatmap. Protein expression changes were normalized to Vinculin (VCL) and Control cell line. **f** Dose-response curve of VCaP cells overexpressing the indicated SPOP species and treated with the ETS-inhibitor YK-4-279. All error bars, mean + s.e.m. P-values were determined by one-way ANOVA (**b**) or two-way ANOVA (**c, e, f**) with multiple comparisons and adjusted using Benjamini-Hochberg post-test ****P < 0.0001. Molecular weights are indicated in kilodaltons (kDa). Source data are provided as a Source Data File.

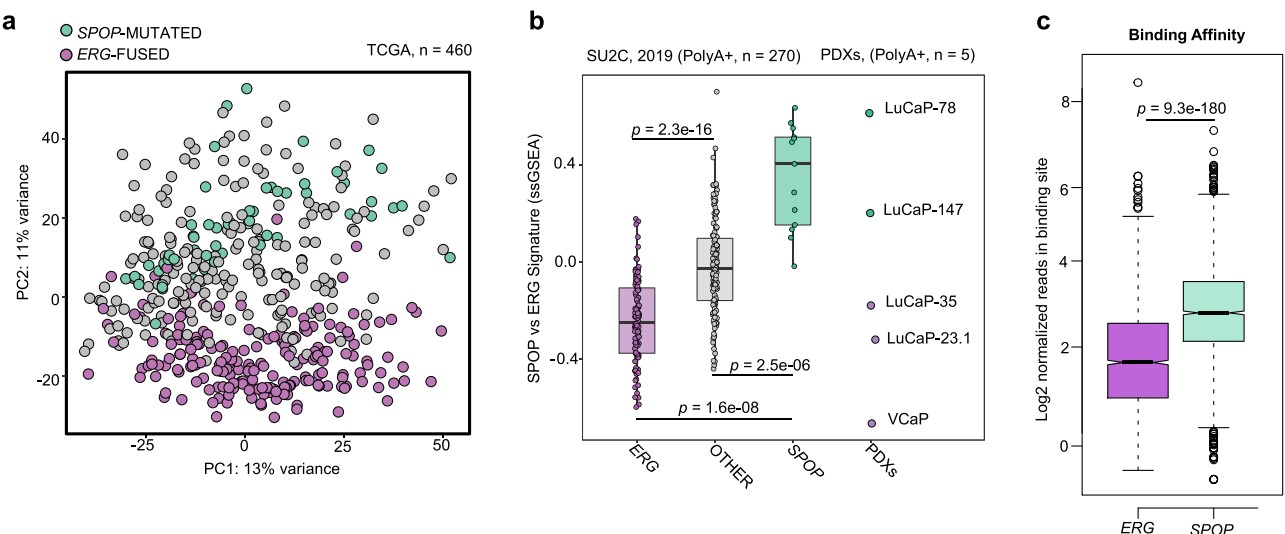

**Fig. 2 Transcriptome analysis of primary prostate cancer patients reveals major differences between *SPOP* mutant and *ERG*-fused tumor subtypes. a** PCA-analysis based on RNA-Seq derived mRNA expression levels (TCGA cohort). *ERG*-fused (violet) and *SPOP*-mutant (green). Individuals were annotated into subtypes as described in Material and Methods. **b** Boxplots representing the transcriptional activity of SPOP integrated-signature (see Materials and Methods) applied to CRPC samples (SU2C-2019 cohort, left) and PDX-models. Scores are determined genes-signatures derived from primary prostate tumors (TCGA-cohort). ERG-fused samples are depicted in violet, SPOP-mutant samples are depicted in green. Samples not harboring SPOP mutations or ERG rearrangements are represented in gray. P-values were determined using Wilcoxon rank sum test and adjusted for multiple testing. Individual. (Boxplots, ERG: max=0.22; min = −0.75; center = −0.31; Q2 (25%) = −0.47; Q3 (75%) = −0.13, OTHER: max = 0.88; min = −0.55; center = −0.03; Q2 (25%) = −0.20; Q3 (75%) = 0.12, SPOP: max = 0.79; min = −0.02; center = 0.50; Q2 (25%) = 0.19; Q3 (75%) = 0.64.) **c** Boxplots depicting the number of normalized reads per binding site across all the differentially bound (DB) regions resulting from the comparison between *ERG*-fused (violet) and *SPOP*-mutant (green) samples. Analysis was restricted to DB regions showing an adjusted P-value (FDR) lower than 0.05 (as determined by DiffBind/DESeq2 pipeline). Subsequently, significance between ERG and SPOP subgroups was assessed using Wilcoxon rank sum test. (Boxplots, ERG: max = 8.14; min = −0.85; center = 1.51; Q2 (25%) = 0.74; Q3 (75%) = 2.39, SPOP: max = 7.03; min = −1.07; center = 2.76; Q2 (25%) = 2.11; Q3 (75%) = 3.48).

androgen receptor (AR) signaling in *SPOP*-mutant vs. *ERG*-fused cancers might be at the root of the incompatibility between the driver events. Thus, we analyzed, in particular, AR- and ERG-related transcription in VCaP cells and generated custom signatures using ChIP-seq data and matched RNA-seq samples (Supplementary Data 3)[27]. As expected, SPOP-MTs increased the transcription of genes bound by AR and induced by its ligand dihydrotestosterone (DHT), whereas genes bound by AR and repressed by DHT were further reduced (Fig. 3a, Supplementary Fig. 5c, and Supplementary Data 3–5). Remarkably, we observed the opposite effect on genes bound only by ERG. Mutant SPOP downregulated ERG-induced genes (e.g., MYC) and upregulated ERG-repressed genes, respectively (Fig. 1d). In line with these findings, gene ontology analysis of AR-ERG co-bound gene signature in VCaP cells indicated that the most striking transcriptional changes were linked to cellular differentiation and cell cycle arrest that are directly induced by DHT and repressed by ERG (e.g., HOXA genes, CDKN1A/p21, Fig. 1d, Fig. 3b, and Supplementary Fig. 5d). To reduce the number of

genes falling within our custom signatures, we used a particularly restrictive approach and considered co-bound only genes where AR and ERG-binding sites overlapped for at least 1 bp. As a result, some genes (e.g., CDKN1A/p21) that are bound both by AR and ERG in their promoter region, but bindings of which do not overlap are not included in this category despite being bona fide co-bound targets.

The dramatic upregulation of this gene set was paralleled by downregulation of cell cycle genes (e.g., E2F and MYC targets), implying a direct link between the induction AR/ERG co-bound genes, the repression of ERG targets, cell differentiation, and the synthetic sick relationship of ERG and mutant SPOP (Fig. 1d, Supplementary Fig. 5a–e, and Supplementary Data 6,7). The relationship of AR- and ERG-related custom signatures to the hallmark gene sets are highlighted in Fig. 3c in a two-dimensional network. Moreover, independently generated signatures of senescence-associated transcripts were enriched in VCaP over-expressing SPOP mutants, further corroborating our data of a senescence-induced cell cycle arrest (Supplementary Fig. 5f)[28,29].

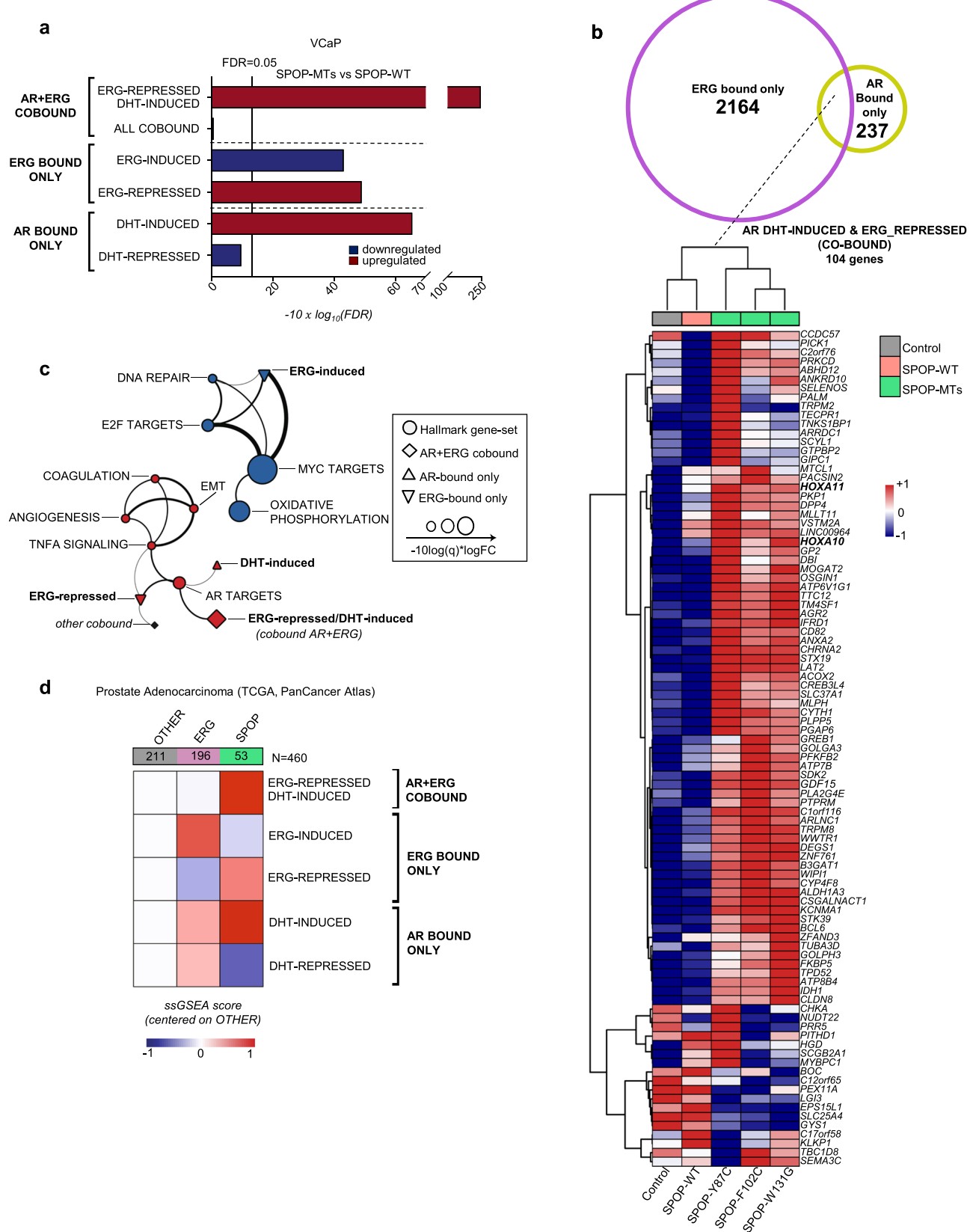

Conversely, we assessed the consequence of ERG overexpression in LNCaP cells under low DHT levels where mutant SPOP triggers AR signaling and tumor growth (Supplementary Fig. 6a, b and Supplementary Data 8 and 9)[19,23]. Over-expression of ΔERG in this setting robustly reverted the induction of signatures related to cell proliferation (e.g., E2F and MYC targets) and AR signaling. Taken together, the data imply a mutual incompatibility of mutant SPOP-induced AR signaling and the function of the ERG oncogene.

Next, we verified if corresponding transcriptional changes were found in clinical tissue samples. Indeed, ERG-regulated genes

**Fig. 3 Mutant SPOP-induced, androgen receptor signaling, antagonizes ERG activity. a** Gene-set enrichment analysis of VCaP cells overexpressing SPOP mutant (SPOP-MTs; SPOP-Y87C, -F102C, -W131G) vs. SPOP-wild type (-WT), based on RNA-seq data. Experiments were performed using three replicates for each condition. Enrichments are determined on custom gene-sets of direct androgen receptor (AR) and ERG target genes (Supplemental Dataset 1). Enrichments and FDR-adjusted *P*-values are computed with *Camera* (pre-ranked) **b** Venn Diagram and heatmap depicting the expression of genes included in the custom gene-set of AR/ERG co-bound genes that are repressed by ERG and induced by DHT in VCaP cells overexpressing SPOP-MTs (SPOP-Y87C, F102C, W131G), SPOP-WT and vector Control. Genes (rows) and samples (columns) were clustered using Euclidean distance. Gene expression values were normalized using variance stabilizing transformation (vst) and subsequently scaled and centered by row prior of clustering. Columns represent the average expression of three replicates for each condition. **c** Two-dimensional network representing overlaps between the ten most significantly enriched Hallmark and custom gene-sets, identified when comparing SPOP-MTs (SPOP-Y87C, F102C, W131G) to SPOP-wild type (-WT) overexpressing VCaP cells. The thickness of edges is proportional to the significance of the overlap of the connected nodes measured by the Fisher test. Only edges with FDR value <0.05 are shown. The size of nodes is proportional to gene-set enrichment significance and equals to $-10 \times \log_{10}$ (FDR). **d** Heatmap representing gene-set activity stratified according to tumor subtype, derived from TCGA cohort. For each tumor group, the average value of single-sample GSEA scores was considered. Values were scaled and referenced to samples that did not harbor any ETS-fusion (*ERG, ETV1,* and *ETV4*) or point mutations in *SPOP*[3].

culled from VCaP cells were upregulated in *ERG*-fused and downregulated in *SPOP*-mutant primary tumors (Fig. 3d, Supplementary Fig. 6c, and Supplementary Data 10, 11)[3]. Notably, the most striking changes between the two groups were found again in the AR/ERG co-bound gene set in primary prostate cancers (Supplementary Fig. 6d and Supplementary Data 12, 13)[3,6]. The results underscore both the relevance of our cell culture-based data and highlight the transcriptional differences among *ERG*- and *SPOP*-driven tumors.

**ZMYND11 is a de novo SPOP substrate**. Using tandem mass tag (TMT)-based quantitative mass spectrometry, we set out to search for SPOP substrates that may influence the activity of AR and ERG and thereby may cause to the synthetic sick relationship between mutant SPOP and ERG in VCaP cells overexpressing mutant SPOP (SPOP-MTs; SPOP-Y87C, -F102C, -W131G, Fig. 4a and Supplementary Data 14). As recurrent loss-of-function SPOP mutants impair substrate ubiquitylation and proteasomal degradation, we searched for proteins the expression levels of which increase without a concomitant increase in mRNA levels (Fig. 4b, Supplementary Fig. 7a, and Supplementary Data 15). Overall, we noted a strong correlation of protein with mRNA expression changes with consistent changes of our AR and ERG custom signatures at the protein level (Fig. 4b and Supplementary Fig. 7b, c). In addition, we found a marked upregulation of the known SPOP substrate and AR activator TRIM24 at the protein level (Figs. 1d and 4a; Supplementary Data 14)[22,23] and subsequently assessed if TRIM24 and more generally AR is implicated in the synthetic sick relationship between mutant SPOP and ERG. Indeed, knockdown of TRIM24 by two short-hairpin RNAs partially reverted the growth inhibition mediated by mutant-SPOP in VCaP cells and reduced AR signaling (Supplementary Fig. 7d–f), while over-expression of AR was sufficient to decrease cellular growth (Supplementary Fig. 7g, h).

The most striking upregulation was noted for the bromodomain histone reader ZMYND11 (Fig. 4b). In line with a SPOP substrate, wild type SPOP bound and decreased the expression of HA-ZYMND11 in a proteasome-dependent manner (Fig. 4c, d). We found two degron sequences that were required for efficient SPOP-mediated ubiquitylation and protein degradation (Fig. 4e±g and Supplementary Fig. 8a.). As expected, SPOP mutants failed to bind and adequately ubiquitylate HA-ZMNYD11-WT (Fig. 4h–j)[10,11,20–22,24–26]. Finally, we confirmed that the expression of mutant SPOP prolonged the half-life of endogenous ZMYND11 in VCaP cells and upregulated ZMYND11 expression in other prostate cancer cells (Fig. 4k and Supplementary Fig. 8b).

**ZMYND11 induces AR signaling pathway and represses ERG activity**. Next, we assessed if ZMYND11 protein upregulation

also contributed to the synthetic sick relationship. In support, forced expression of the degron-deficient variants of ZMYND11 (HA-ZMYND11-DMT1/DMT2) was sufficient to diminish the growth of VCaP cells (Fig. 5a, b and Supplementary Fig. 8c), while knockdown of ZMYND11 partially reverted the growth inhibition mediated by mutant SPOP (Fig. 5c).

We postulated that ZMYND11 upregulation could contribute to the synthetic sick relationship by repressing the ERG oncogene's transcriptional activity or enhancing AR signaling. To this end, expression changes induced by HA-ZMYND11-DMT2 largely overlapped with genes perturbed by mutant SPOP while the opposite was noted when ZMYND11 expression was reduced by RNA interference (Fig. 5d, Supplementary Fig. 8d, and Supplementary Data 16). In comparison to mutant SPOP, AR and ERG target genes were similarly dysregulated by HA-ZMYND11-DMT2 (Fig. 5e and Supplementary Data 17, 18). As the PWWP domain of ZMYND11 has been involved in the regulation of transcription through its ability to bind H3K36me3 histone marks[30], we tested this domain's contribution to the overall transcriptional output. Indeed, the PCA of VCaP cells over-expressing either HA-ZMYND11-DMT2 or a PWWP domain deficient mutant (W294A) revealed a major contribution of this domain to the ZMYND11 induced transcriptional changes (Supplementary Fig. 8e).

Subsequently, we mapped the genomic occupancy of ZMYND11 in VCaP cells expressing the SPOP-Y87C mutant by chromatin immunoprecipitation sequencing (ChIP-seq) and found an enrichment of ZMYND11-binding sites at promoter regions controlling ERG-induced genes (e.g., MYC,) and AR/ERG co-bound genes (e.g., p21/CDKN1A) (Fig. 5f, g, Supplementary Fig. 9a–e, and Supplementary Data 19–22). The data imply a critical enhancer function of ZMYND11 in boosting AR signaling and repressing ERG signaling downstream of mutant SPOP.

**Wild type SPOP is required for ERG oncogenic function**. We reasoned that ERG-driven tumors might require wild type SPOP to degrade ZMYND11 and unlock ERG's oncogenic function. In support, over-expression of wild type SPOP increased the 3D growth of mouse prostate epithelial organoids and VCaP cells only when ERG was over-expressed (Fig. 1b, Supplementary Figs. 2c and 10a, b). Remarkably, ERG-fused human tumor tissues also displayed the highest SPOP mRNAs levels (Fig. 6a). Thus, we wondered if ERG itself may directly upregulate SPOP transcription to support its own oncogenic activity. Indeed, mining ERG ChIP-seq data in VCaP cells revealed ERG bindings sites in the promoter region of SPOP (Supplementary Fig. 10c). Moreover, knockdown of ERG reduced SPOP protein levels in VCaP cells, while forced expression of a ΔERG led to the upregulation of SPOP mRNA and protein levels in PC3 cells (Fig. 6b, Supplementary Figs. 3e and 10d).

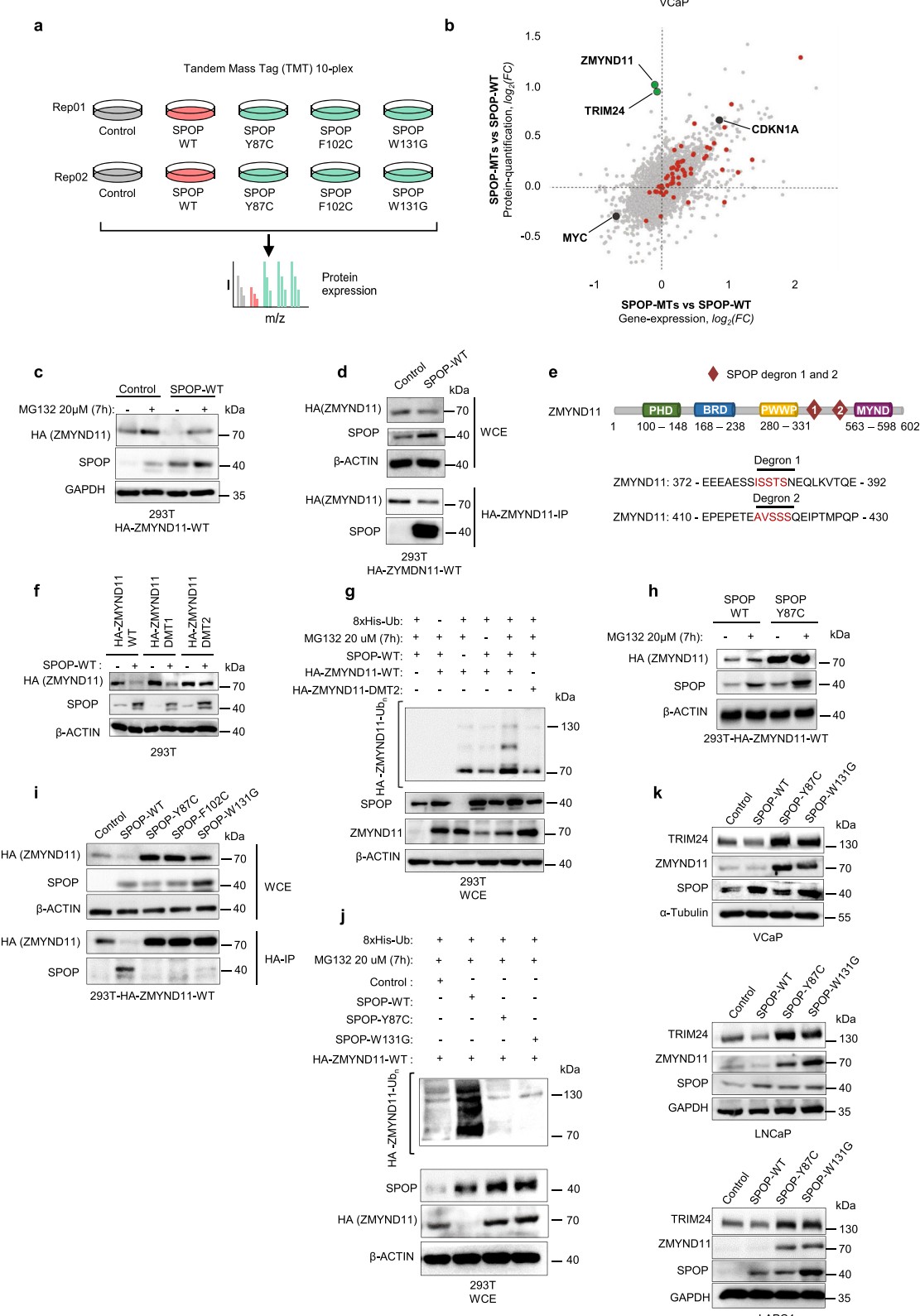

We then asked if the elevated SPOP levels in the context of forced ΔERG expression have a functional impact on the oncogenic activity of ΔERG in the androgen-independent PC3 cells, in which ERG promotes tumor cell invasion[31]. Indeed, the reduction of SPOP levels by RNA interference reduced the ability of ΔERG to invade into Matrigel (Fig. 6c). Similarly, knockdown of SPOP in VCaP cells reduced cell growth in 3D cell culture and

impaired ERG-mediated gene transcription (Supplementary Fig. 10e, f and Supplementary Data 23 and 24). In accordance with the ability of mutant SPOP to repress the function of endogenous wild type SPOP in a dominant-negative manner, the over-expression of mutant SPOP (SPOP-Y87C, -F102C, -W131G, -F133S) phenocopied the effect of SPOP knockdown on ERG-mediated invasion in PC3 cells (Supplementary Fig. 10g, h). In

**Fig. 4 ZMYND11 is a de novo SPOP substrate. a** Schematic illustration showing the design of the proteomics experiments. Tandem Mass Tag (TMT)-based quantitative mass spectrometry ($n = 2$) was used in VCaP cells overexpressing Control vector (Control), SPOP-WT, or three different SPOP mutants (SPOP-Y87C, SPOP-F102C, and SPOP-W131G). **b** Scatter plot comparing transcriptomic and proteomic derived fold changes resulting from the comparison between SPOP-MTs (average across SPOP-Y87C, -F102C, -W131G) and SPOP-WT VCaP cells. ($n = 3$ for transcriptome, $n = 2$ for proteome). Genes belonging to DHT-Induced/ERG-Repressed gene-signature (AR + ERG co-bound) are highlighted in red. TRIM24 and ZMYND11 are the most upregulated proteins without changes at mRNA levels and are highlighted in green. CDKN1A (p21) upregulated at both mRNA and protein levels, and MYC downregulated at both mRNA and protein levels are highlighted in black. **c** Over-expression of HA-ZMYND11 and SPOP-WT in 293T cells and subsequent expression analysis of the indicated proteins by immunoblotting ($n = 2$). **d** Whole-cell extracts (WCE) of 293T cells over-expressing HA-ZMYND11-WT and different SPOP species and corresponding anti-HA-immunoprecipitation (HA-IP). Expression of the indicated proteins was analyzed by immunoblotting ($n = 1$). **e** Domain structure of ZMYND11 with indicated SPOP-degron and ubiquitin sites. **f** Forced expression of SPOP-WT together with HA-ZMYND11-WT or two degron-deficient mutants (DMT1 & DMT2) in 293T cells ($n = 1$). **g** In vivo ubiquitylation assay of HA-ZMYND11 in 293T cells. Cells were transiently transfected with the indicated constructs, and histidine-tagged (his-tag), ubiquitylated proteins were pulled down using nickel beads. Ubiquitylated HA-tagged ZMYND11 was detected by immunoblotting ($n = 1$). **h** Over-expression of HA-ZMYND11 and SPOP-Y87C in 293T cells and subsequent expression analysis of the indicated proteins by immunoblotting after proteasomal inhibition with MG132. **i** Whole-cell extracts (WCE) and corresponding anti-HA-immunoprecipitation (HA-IP) of 293T cells over-expressing HA-ZMYND11-WT and different SPOP-MTs species as indicated. Expression of the indicated proteins was analyzed by immunoblotting ($n = 1$). **j** In vivo ubiquitylation assay of HA-ZMYND11 in 293T cells. Cells were transiently transfected with the indicated constructs and histidine-tagged (His-tag); ubiquitylated proteins were pulled down using nickel beads. Ubiquitylated HA-tagged ZMYND11 was detected by immunoblotting ($n = 1$). **k** Immunoblots of indicated proteins in VCaP, LNCaP, and LAPC4 human prostate cancer cells overexpressing the indicated SPOP species. Molecular weights are indicated in kilodaltons (kDa) ($n = 3$). Source data are provided as a Source Data File.

agreement with the established repressive function of ZMYND11 on ERG, we found that over-expression of HA-ZMYND11-DM2 was sufficient to repress ERG-induced invasion and established target genes in PC3 cells (Fig. 6d, e). Taken together, the data imply the existence of a positive feed-forward loop, in which ΔERG promotes the expression of SPOP to sustain its oncogenic activity.

**ERG and mutant SPOP trigger different responses to therapeutic interventions**. Based on the differences mentioned above in tumorigenesis, we speculated that ERG or mutant SPOP could also trigger different therapeutic responses. In light of the dependency of ERG-driven tumors on wild type SPOP function, we hypothesized that ERG-fusion-positive cells might be particularly sensitive to pharmacological inhibition of SPOP. We analyzed the response of the SPOP small molecule inhibitor compound 6b (SPOP-i) in ERG-fused, SPOP mutant, and other prostate cancer cell lines and patient-derived xenograft models (PDX)[18]. The SPOP inhibitor increased the protein but not the mRNA levels of established SPOP substrates and ZMYND11, while the related inactive analog compound 6c did not (Supplementary Fig. 11a–c). The latter did also not exert any activity in 3D culture models (Supplementary Fig. 11d). In agreement with our previous results, we found that ERG-fused cells (VCaP, LuCaP-23.1, −35) were more sensitive to SPOP-i than ERG-negative cells (22Rv1, LNCaP, PC3), while SPOP mutant cells (LuCaP-78, −147) were particularly insensitive in 3D culture models and in xenograft tumor models in vivo (Fig. 7a–f and Supplementary Fig. 11e). We further validated our results in the mouse prostate epithelial organoids and confirmed the increased sensitivity of ΔERG-expressing cells to SPOP inhibition in this isogenic system (Fig. 7g).

Given the notion that wild type SPOP dampens AR function in the context of ERG to sustain tumor growth, we asked if VCaP cells are particularly susceptible to increased DHT levels. Indeed, exposure to a high-dose of testosterone in vivo or DHT in vitro induced similar molecular changes as for the over-expression of mutant SPOP and greatly suppressed the growth of ERG-fusion-positive cells but not of SPOP mutant cells in vitro and in vivo (Figs. 1c and 8a–f; Supplementary Fig. 12a–e and Supplementary Data 25 and 26). Moreover, signatures of senescence-associated transcripts were also enriched in VCaP cells upon treatment with DHT, further corroborating our data of a senescence-induced cell

cycle arrest (Supplementary Fig. 12f). Strikingly, the sensitivity to SPOP-i and to high testosterone in vivo correlated well with ERG protein expression levels in the respective ERG-fusion-positive cell line and PDX model (Fig. 8g, h). The data suggest a therapeutic opportunity for SPOP inhibition or high-dose androgen therapy in prostate cancers that express high levels of ERG.

Conversely, and because SPOP mutant cancers are driven predominantly by androgen signaling and consequently display high-level activation of AR-related transcripts in human tumor tissues, we speculated that these tumors might be particularly susceptible to androgen deprivation or antiandrogen therapies (ADT) (Supplementary Fig. 4c). Indeed, the prevalence of SPOP mutations in primary tumors -and tumors that had progressed after initial surgery or radiotherapy- is consistently higher as compared to tumors that had become resistant to subsequent ADT (also referred to as castration-resistant prostate cancer, CRPC, Supplementary Fig. 13a). In line with the notion that this difference may be related to a better response of SPOP mutant tumors to ADT, SPOP mutant tumors display a trend towards better overall survival despite progressing faster after initial therapy (Fig. 9a, b). To functionally analyze androgen deprivation or the antiandrogen enzalutamide response, we chose to ectopically express different SPOP variants and ΔERG in the androgen-dependent human LAPC4 prostate cancer cells that are wild-type for both driver genes. In accordance with the clinical observation, the presence of mutant SPOP (SPOP-Y87C, SPOP-W131G) rendered LAPC4 cells more susceptible to either ADT or enzalutamide in comparison to cells expressing control vector (Fig. 9c and Supplementary Fig. 13b).

In contrast, ΔERG rendered the same cells more resistant to enzalutamide. In line with the previous findings in VCaP and LuCaP-147 cells, ΔERG expression rendered LAPC4 cells susceptible to high levels of DHT, while mutant SPOP had the opposite effect (Supplementary Fig. 13b). Taken together, the different responses to established and experimental therapeutic modalities observed between mutant SPOP and ERG add further credence to their divergent roles of the AR pathway related to tumorigenesis.

## Discussion

Although multiple studies over recent years have uncovered different genetically-defined subtypes of primary prostate cancer,

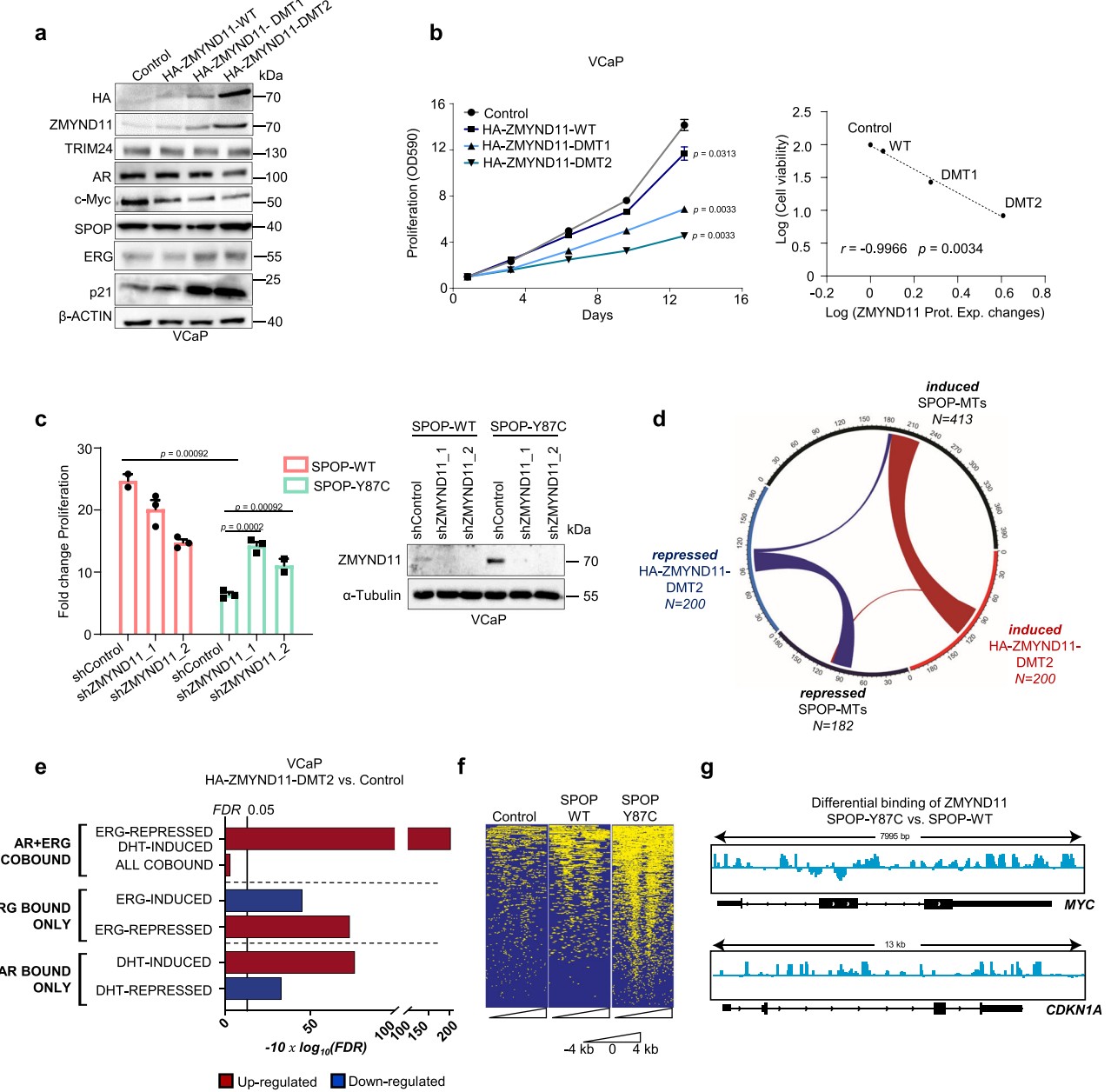

**Fig. 5 ZMYND11 induces AR signaling pathway and represses ERG activity. a**, **b** Immunoblot of indicated proteins ($n = 2$) (**a**) and corresponding 2D proliferation assay (**b**) of VCaP cancer cells overexpressing HA-ZMNYD11-WT and derived degron-deficient mutants (DMT1/2) ($n = 3$). Correlation between cell viability and ZMYND11 protein expression changes (Prot. Exp. Changes), as quantified by immunoblot in the same cell lines. *P*-values were calculated using the Pearson rank correlation. **c** Fold-change cell viability of VCaP cancer cells overexpressing the indicated SPOP species with and without *ZMYND11* knockdown using two different short-hairpin RNAs, at day 16 ($n = 3$) one-way ANOVA with multiple comparisons and adjusted using Benjamini–Hochberg post-test. The protein expression of the indicated proteins was analyzed by immunoblotting. **d** Chord-diagram of transcriptionally regulated genes by either SPOP-MTs or HA-ZMYND11-DMT2 in VCaP cells (FDR < 0.05). Strings, whose thickness is proportional to the number of shared elements, represent common genes between sets. **e** Gene-set **e**nrichment analysis of VCaP cell overexpressing HA-ZMYND11-DMT2 compared to Control, based on RNA-seq data. Enrichments are performed on custom gene-sets of direct androgen receptor (AR) and ERG target genes. FDR-adjusted *P*-values are computed with *Camera* (pre-ranked). **f** Heatmap of**f** ChIP-seq signals around TSS regions ($+/-4$ kb) at which ZMYND11 bindings were identified by peak calling procedure (Macs2) in VCaP cells overexpressing the indication constructs. **g** IGV-derived screenshots representing loglikelihood ratio of ZMYND11 bindings in mutant SPOP (SPOP-Y87C) vs. wild-type SPOP over-expressing VCaP cells. Reported are *MYC* (up) and *CDKN1A* (bottom). All error bars, mean ± s.e.m. *P*-values were determined by two-way ANOVA (**b** Molecular weights are indicated in kilodaltons (kDa). Source data are provided as a Source Data File.

their biological understanding and therapeutic implications remain a largely unexplored territory. Here, we report two diametrically different paths toward tumorigenesis triggered by either highly recurrent missense mutation in *SPOP* or gene fusion involving the *ERG* oncogene. Importantly, wild-type SPOP emerges as a critical component that enforces oncogenic ERG signaling in part through dampening AR activity, while mutant SPOP drives tumorigenesis through activation of AR signaling. Moreover, several studies have previously highlighted the importance of AR target genes in the context of SPOP mutants

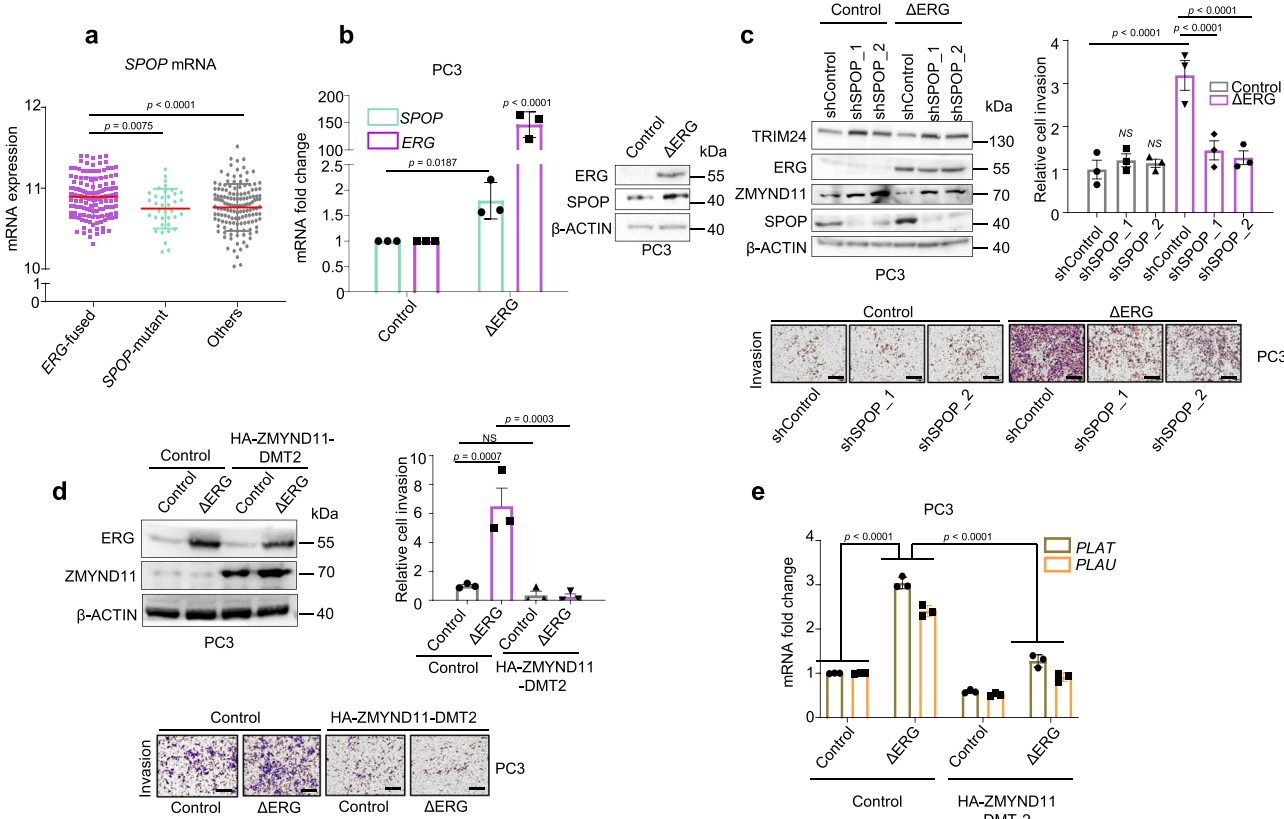

**Fig. 6 SPOP-WT is an ERG target gene and required for ERG-mediated cell invasion. a** *SPOP* mRNA expression levels in 333 primary prostate cancer tissues stratified according to the indicated driver mutations[3]. Error bars, mean ± s.d. **b** SPOP mRNA and protein levels in response to forced expression of ΔERG in PC3 prostate cancer cells by qPCR and immunoblotting, respectively. Error bars, mean + s.e.m. ($n = 3$). *P*-values were determined by unpaired, two-tailed Student's *t*-test. Control vs. ΔERG for *SPOP* expression levels. Control vs. ΔERG for *ERG* expression levels. **c** Transwell Matrigel invasion assay of PC3 cells with forced expression of ΔERG and knockdown of SPOP using two different short-hairpin RNAs. Protein expression of the indicated proteins was assessed in parallel by immunoblotting. Error bars, mean ± s.e.m. ($n = 3$) (bar represents 400 μm). **d** Transwell Matrigel invasion assay of PC3 cells with forced expression of ΔERG and HA-ZMYND11-DMT2 and corresponding immunoblot analysis. Error bars, mean ± s.e.m. ($n = 3$) (bar represents 400 μm). **e** Analysis of the ΔERG- and HA-ZMYND11-DMT2-induced transcriptional changes in the ERG target genes *PLAU* and *PLAT*. All error bars, mean ± s.e.m. *P*-values were determined by one-way ANOVA with multiple comparisons and adjusted using Benjamini–Hochberg post-test (**a**, **c**, **d**, **e**). NS, not significant. Molecular weights are indicated in kilodaltons (kDa). Source data are provided as a Source Data File.

and ERG-positive tumors[19,32,33]. Based on the incompatibility between the two tumor subtypes, our work enabled the development of specific custom signatures related to AR and ERG transcript that are necessary to drive proliferation and tumorigenesis in the context of *ERG*-positive and *SPOP*-mutants tumors. In addition, we show that the bromodomain histone reader ZMYND11 is a SPOP substrate implicated downstream of SPOP in the opposing regulation of the ERG and AR pathway in the two tumor subtypes (Fig. 10). The AR and ERG pathways have been previously reported to have a partially antagonistic relationship[33,34], further corroborating our findings.

As activation of the androgen receptor by androgens represents a key lineage specific oncogenic pathway in prostate cancer, androgen deprivation/antagonization therapies (ADT) remain the uniform treatment modality up to this very day. That said, the responses to ADT are highly variable and may last from a few weeks up to many years. Here, we provide functional evidence that pre-existing prostate cancer founder mutations influence the treatment response. Most notably, *SPOP* mutations promote susceptibility to androgen deprivation therapies. In agreement with our findings, earlier reports have shown the underrepresentation of *SPOP* mutant tumors in cohorts of castration-resistant disease and a more favorable response to the abiraterone and enzalutamide[35,36].

Conversely, we show that the ERG oncogene's presence increases the susceptibility of tumor cells to high-dose androgen therapy, while cells expressing mutant SPOP remain largely unaffected. This is of clinical interest because testosterone treatment of patients with the advanced castration-resistant disease has recently shown to trigger anti-tumor responses in around one-third of the patients[37]. It is tempting to speculate that these insights may help to discern responders from non-responders.

In addition, we provide evidence that the antagonistic relationship between mutant SPOP and ERG may be used towards the development of new therapeutic avenues. More specifically, we show that ERG-driven cancer cells are particularly sensitive to the inhibition of wild-type SPOP using recently developed small molecule inhibitors[18]. Our preclinical data suggest that SPOP inhibition may be effective in clinical settings where ERG is robustly expressed (e.g., neo-adjuvant setting or early metastatic disease).

Our results generally identify another paradigm for antagonistic driver genes in prostate cancer that has recently emerged for other cancer types[38–40]. In analogy to prostate cancer, truncal point mutations in *DNMT3A* and gene fusions in *PML-RARA* are mutually exclusive drivers in acute myeloid leukemia (AML). Similar to SPOP, intact DNMT3A has been found to be critical for PML-RARA-driven leukemia (Supplementary Fig. 14a, b)[41,42].

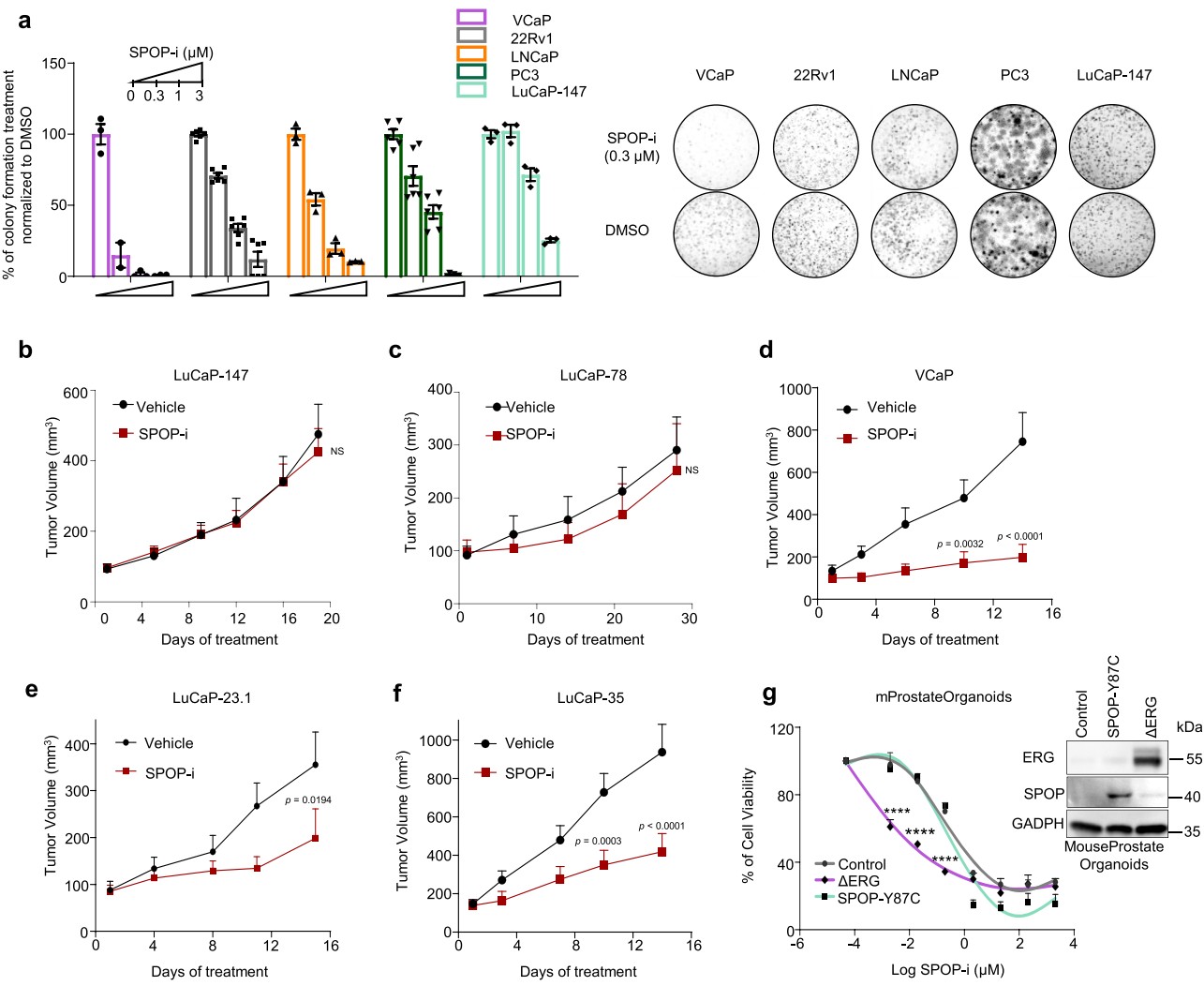

**Fig. 7 ERG-positive tumor cells are particularly sensitive to SPOP inhibition. a** SPOP inhibitor (SPOP-i, compound 6b) mediated 3D growth inhibition in methylcellulose in the indicated prostate cancer cell lines. **b** Tumor growth kinetics with ($n = 10$) or without (vehicle; $n = 10$) SPOP-i treatment in xenografts established from LuCaP-147 (SPOP-Y83C) PDX cells. **c** Tumor growth kinetics with ($n = 4$) or without (vehicle; $n = 4$) SPOP-i treatment in xenografts established from LuCaP-78 (SPOP-W131G) PDX cells. **d** Tumor growth kinetics with ($n = 11$) or without (vehicle; $n = 11$) SPOP-i treatment in xenografts established from VCaP. **e** Tumor growth kinetics with ($n = 6$) or without (vehicle; $n = 8$) SPOP-i treatment in LuCaP-23.1 (ERG-positive)
**f** Tumor growth kinetics with ($n = 8$) or without (vehicle; $n = 10$) SPOP-i treatment in LuCaP-35 (ERG-positive) PDX PDX. All SPOP-i treatment initiated when tumors reached an average of 100 mm³. **g** Dose–response curves to SPOP-i treatment of Mouse Prostate Organoids overexpressing ΔERG, SPOP-Y87C and Control vector. All error bars, mean + s.e.m. P-values were determined by two-way ANOVA with multiple comparisons and adjusted using Šidák post-test. NS, not significant. ****$P < 0.0001$. Molecular weights are indicated in kilodaltons (kDa). Source data are provided as a Source Data File.

Importantly, we demonstrate here for prostate cancer that the concept of antagonistic driver genes can be exploited to identify therapeutic opportunities.

## Methods

**Cell culture, transfection, and infection.** VCaP, LNCaP, PC3, 22Rv1, HEK293 cells were purchased from ATCC. LAPC-4 was a gift from Prof. Helmut Klocker. VCaP and HEK293 were grown in DMEM with Glutamax (Gibco); LNCaP, PC3, 22Rv1, LAPC-4 in RPMI medium (Gibco); all were supplemented with 10% full bovine serum (FBS; Invitrogen), or 10% charcoal-stripped serum (CSS; One-Shot Fetal Bovine Serum, Charcoal Stripped, Gibco) for androgen deprivation therapy response, and 1% Penicillin/Streptomycin. LuCaP-147 were grown in StemPro medium (hESC SFM StemPro, Gibco) with regular supplements. All cells were incubated at 37 °C and 5% CO₂ and routinely tested for mycoplasma contamination.

For stable knockdown experiments, cells were infected with pLKO-1 vectors (Sigma) and the following clones were used; *SPOP*: TRCN0000140431 (shSPOP_1) and TRCN000013911 (shSPOP_2); *TRIM24*: TRCN000021262 (shTRIM24_1) and TRCN0000195528 (shTRIM24_2); *ERG*: TRCN0000429354 (shERG_1) and TRCN0000432394 (shERG_2); *ZMYND11*: TRCN0000275479 (shZMYND11_1)

and TRCN0000275542 (shZMYND11_2). After infection, cells were selected in the presence of puromycin (2 µg/ml).

For SPOP, ΔERG, HA-ZMYND11-WT, HA-ZMYND11-DMT1, HA-ZMYND11-DMT2, MYC, and AR over-expression a derivate of the pLX304 vector was used throughout in which the CMV promoter has been exchanged to a PGK promoter, and the blasticidin cassette left unchanged (ΔERG constructs) or exchanged by a puromycin resistance cassette (SPOP constructs) (pLX_TRC_307, available at Addgene as Plasmid 41392, pCW107). All ORFs were cloned into pLX_TRC_307 using Nhe1 and Mlu1. Tumors from PDX LuCaP-78, −147, −35, −23.1 were collected, dissociated and cultured as previously described[43]. Briefly, PDX tumors were collected in phosphate-buffered saline (PBS) and dissociated on a petri dish in Advanced DMEM (Hepes, Glutamine, Pen/Strep) with 200 U/ml of collagenase Type I (Cat#SCR103) followed by 45 min of incubation at 37 °C. Next, the cell suspension was filtered using a cell strainer 100 µM and centrifuged at 300 × g for 5 min. Red blood lysis buffer (Cat# 11814389001) was added to the solution for 5 min to eliminate red blood cells. Finally, the dissociated tumor was resuspended in PBS and used accordingly for cell culture.

**Chemicals.** MG-132 (M7449) and Cycloheximide (CHX, C4859) were purchased from Sigma and used at 20 µM and 100 µg/ml in all experiments, respectively. SPOP inhibitor (SPOP-i, compound 6b) and its inactive analog (compound 6c),

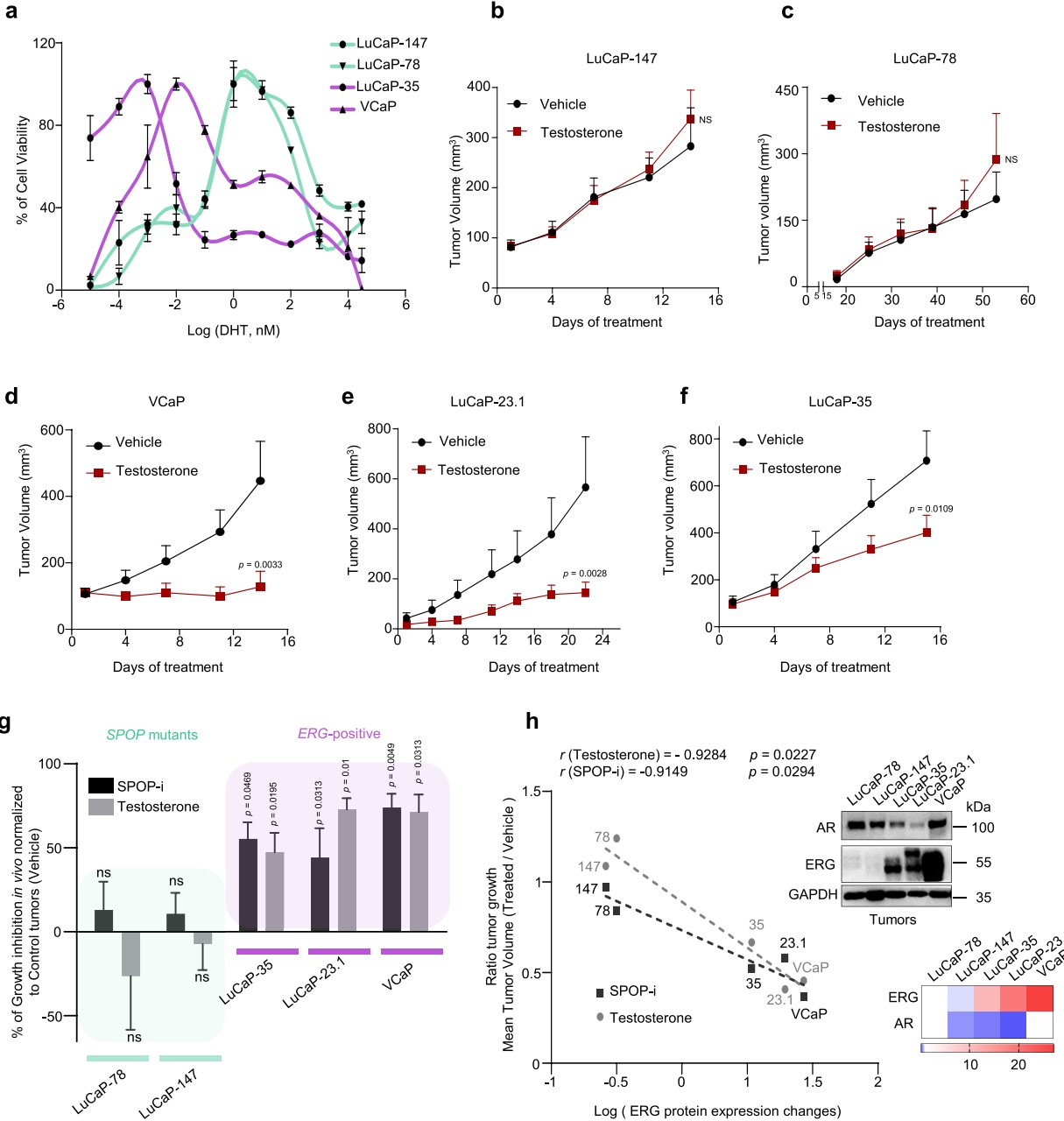

**Fig. 8 ERG and mutant SPOP trigger different responses to therapeutic interventions. a** Dose–response curve to DHT treatment of VCaP, LuCaP-35, LuCaP-78, and LuCaP-147 PDX cancer cells. Before DHT treatment, PDX were grown in standard media without DHT. VCaP were starved for 24 h in CSS medium (RPMI + 10% charcoal-stripped serum). Cell viability was assessed after 2 weeks. **b** Tumor growth kinetics with ($n = 10$) or without (vehicle; $n = 10$) testosterone treatment in xenografts established from LuCaP-147 (SPOP-Y83C). **c** Tumor growth kinetics with ($n = 4$) or without (vehicle; $n = 4$) testosterone treatment in xenografts established from LuCaP-78 (SPOP-W131G) cells. **d** Tumor growth kinetics with ($n = 6$) or without (vehicle; $n = 10$) testosterone treatment in xenografts established from VCaP (ERG-positive) cells. **e** Tumor growth kinetics with ($n = 12$) or without (vehicle; $n = 12$) testosterone treatment in xenografts established from LuCaP-23.1 (ERG-positive) cells. **f** Tumor growth kinetics with ($n = 10$) or without (vehicle; $n = 10$) testosterone treatment in xenografts established from LuCaP-35 (ERG-positive) cells. **g** Sensitivity to Testosterone and SPOP-i treatment in xenograft and PDX models. LuCaP-23.1, LuCaP-35 and VCaP are *ERG*-positive cancer cells. LuCaP-147 and LuCaP-78 are *SPOP* mutant cancer cells (respectively SPOP-Y83C and SPOP-W131G). Growth inhibition is calculated using the last tumor measurements as shown in b-f and Fig. 7b–f. **h** Correlation of sensitivity to SPOP-i or testosterone treatment shown in Extended Data Figs. 8f–j and 9e–i, with ERG protein levels, as quantified by immunoblot, in PDX models and xenografts. *P*-values were calculated using Pearson rank correlation. Corresponding immunoblot and quantification of AR and ERG protein levels depicted as a heatmap. Protein expression changes were normalized to GAPDH and LuCaP-78. All error bars, mean + s.e.m. *P*-values were determined by two-way ANOVA with multiple comparisons and adjusted using Benjamini–Hochberg post-test (**b–e**) or by unpaired, two-tailed Student's *t*-test (**g**), NS, not significant. Molecular weights are indicated in kilodaltons (kDa). Source data are provided as a Source Data File.

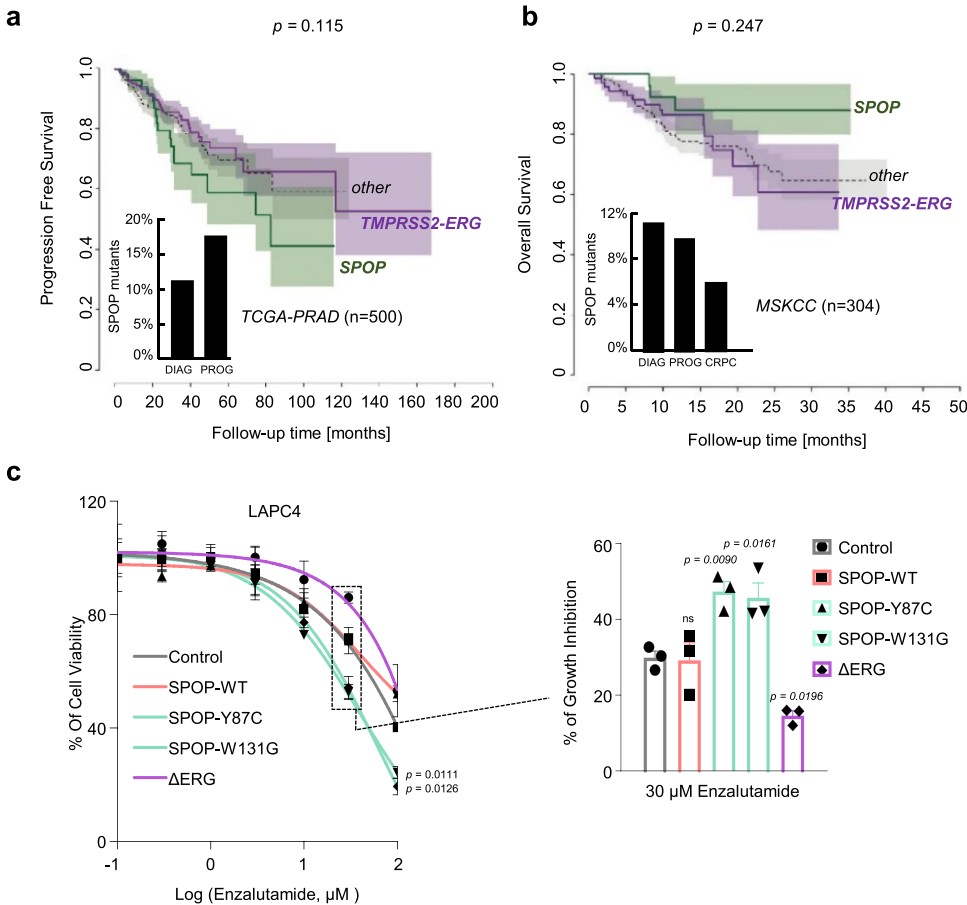

**Fig. 9 *SPOP* mutant tumors are particularly susceptible to androgen deprivation therapies (ADT). a** Progression-free survival of prostate cancer patients derived from the TCGA-cohort. Curves representing TMPRSS2-ERG rearranged, and SPOP-mutant patients are indicated in violet and green, respectively. The area around the curves represents 80% confidence interval. The bar plot in the lower left corner indicates the percentage of SPOP-mutant tumors within all patients diagnosed with prostate cancer (DIAG) and within the individuals who developed a progression of the disease (PROG). *P*-values for Kaplan–Meier curves were determined using the log-rank test ($P = 0.115$). **b** Overall survival of prostate cancer patients derived from the MSK-IMPACT cohort. Curves representing *TMPRSS2-ERG* rearranged, and *SPOP*-mutant patients are indicated in violet and green, respectively. The area around the curves represents 80% confidence interval. The bar plot in the lower left corner indicates the percentage of SPOP-mutant tumors within all patients who were diagnosed with prostate cancer (DIAG), within individuals who developed a metastatic progression of the disease (PROG), and within individuals who developed castration-resistant prostate cancer (CRPC). *P*-values for Kaplan–Meier curves were determined using the log-rank test. ($P = 0.247$). **c** Enzalutamide sensitivity of LAPC4 cells overexpressing ΔERG or SPOP mutant species (Y87C, W131G). All error bars, mean + s.e.m. *P*-values were determined by unpaired, two-tailed Student's *t*-test (**c**), NS, not significant.

were provided by the laboratory of C. Yang (State Key Laboratory of Drug Research, Shanghai Institute of Materia Medica). DHT (5α-Dihydrotestosterone) was purchased from Sigma (D-073), MDV3100 (Enzalutamide) was purchased from APExBIO (A3003). YK-4-279 (ETS inhibitor) was purchased from Selleckchem. All chemicals were used at the indicated concentration.

**Dose–response curves and cell-growth assays**. Cells were seeded (between $1 \times 10^3$ and $1 \times 10^4$ per well) in a 96-well plate. Cells were subsequently treated with serial dilutions of DHT (in 10% CSS medium), or enzalutamide, SPOP inhibitor, ETS inhibitor to determine dose–response curves or were left untreated for cell-growth assays. Proliferation at corresponding time points was assessed by MTT (Methylthiazolyldiphenyl-tetrazolium bromide) assay according to the manufacturer's recommendations (Sigma). For each time point, absorbance (OD, 590 nm) was measured in a microplate reader.

**SA-β-galactosidase staining of VCaP cells**. Senescence-associated-β-galactosidase (SA–β-gal) staining was performed using the Senescence β-Galactosidase Staining Kit (Cell Signaling, #9860) following the manufacturer's instructions. To avoid false-positive staining, we adapted the β-Galactosidase Staining Solution to pH 7. Cells incubated with glycerol 70% were observed under a bright-field microscope.

**Matrigel invasion assay**. Invasion assay was performed as previously described[44]. Briefly, an equal number of PC3 cells were seeded into 10 cm dishes and starved

with a medium without fetal bovine serum for 24 h; subsequently, $1 \times 10^5$ cells were resuspended in 100 μl of starved medium and seeded onto the basement of a Boyden chamber (CLS3422; Sigma) coated with Matrigel. RPMI with 10% fetal bovine serum was added to the lower chamber. After 48 h, invaded cells were fixed with 10% formalin and stained with crystal violet. Absorbance was measured at 560 nm.

**Clonogenic assay in methylcellulose**. Cells were seeded (between $5 \times 10^3$ and $1 \times 10^4$) in methylcellulose (Methocult H4100, StemCell Technologies) in triplicate. Cells were left untreated for cell-growth assay. For SPOP inhibitor assay, cells were treated with vehicle (0.1% DMSO) or drug (SPOP-i) at the corresponding concentration. For androgen therapy, cells were treated with vehicle (0.01% Methanol) or DHT at the corresponding concentration. Cells were incubated at 37 °C and 5% $CO_2$ for 7–28 days, and colonies were stained with MTT solution at 37 °C overnight, and absorbance (OD, 590 nm) was measured in a microplate reader.

**Mouse prostate organoid generation and experiments**. Prostate tissue was extracted from euthanized mice, digested, and seeded in Matrigel as previously described[45]. Briefly, the urogenital system was isolated from the mouse. Seminal vesicles, vas deferens, urethra, and the bladder were removed. The clean prostate was minced in small pieces (1 mm³) and digested in 5 mg/ml of collagenase type II (Collagenase Type II nr. 9001-12-1, gibco) with 10 μM of Y-27631 dihydrochloride for 1–2 h at 37 °C on a moving platform. Later, the minced prostate was centrifuged at $150 \times g$ for 5 min, and the pellet was resuspended in TrypLE (gibco TrypLE

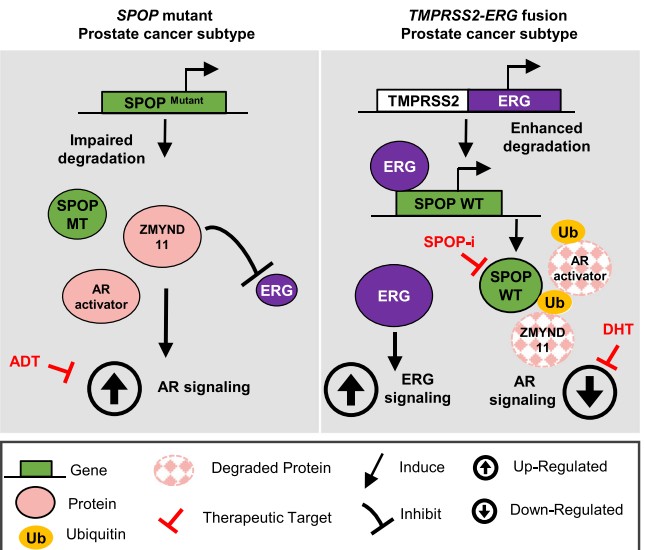

**Fig. 10 Schematic representation of the proposed model for the aversive relationship between mutant *SPOP* and *ERG* in prostate cancer.** On the left side, *SPOP* mutant prostate cancer tumorigenesis is depicted. *SPOP* mutant tumors impair the degradation of substrate proteins such as AR activators (e.g, TRIM24) and ZMIND11, which ultimately triggers the AR pathway and dampens the ERG signaling. In this context, *SPOP* mutant tumors are more susceptible to ADT therapy. On the right side, *TMPRSS2-ERG* mutant tumorigenesis is depicted. ERG binds to the promoter of wild-type SPOP, upregulating its protein expression. Consequently, AR activators and ZMYND11 substrate proteins get degraded, leading to the AR pathway downregulation and unleashing the ERG signaling pathway. *TMPRSS2-ERG* mutant tumors are thus sensitive to high-dose androgen therapy or SPOP inhibition.

Express) with 10 µM of Y-27631 dihydrochloride for 15 min at 37 °C. The dissociated prostate was washed once by topping up to 10 ml with adDMEM/F12 (gibco) and then centrifuged at 150 × g for 5 min. Twenty-thousand cells cells per drop were plated in Matrigel (Corning Matrigel Nr. 356231). To overexpress *SPOP* species and *ΔERG* genes, mouse prostate cells were virally infected by spinoculation for 1 h at 600 × g at 32 °C and selected with puromycin. For the "organoid formation assay," $1.5 \times 10^4$ single cells were plated per well onto 40 µl of Matrigel on day 1 and organoids were grown in "revised human prostate organoids medium" as previously described[43]. Briefly, the medium included the following reagents: adDMEM/F12 (gibco), glutaMAX (2 mM), Pen-strep 100 u/ml, HEPES (10 mM), B27 (1X), EGF (5 ng/ml), AB3-01 (500 nM), Noggin (100 ng/ml), R-Spondin 1 (500 ng/ml), DHT (1 nM), FGF-2 (5 ng/ml), FGF-10 (10 ng/ml), Prostaglandin E2 (1000 nM). The number of formed organoids that reached 100 µM of diameter was counted on days 14 post-plating with cellSens software (Olympus). For the Dose–Response experiment $1 \times 10^4$ mouse prostate cells were plated in 40 µl of Matrigel and treated with vehicle (0.1% DMSO) or drug (SPOP-i) at the indicated concentration for 7 days. Live/dead staining was performed using Calcein AM (final concentration 1 µM) and Ethidium Homodimer I (EthD-1, final concentration (1.33 µM) dissolved in the medium for 1 h. GFP or RFP-positive organoids were analyzed under a fluorescence microscope. The percentage of dead organoids (RFP-positive) was normalized to the Etoposide-treated (positive control) organoids. The genetically engineered Mouse Prostate Organoids were derived from PbCre;R26$^{F133V}$ [13].

**Immunohistochemistry.** Cytoblocks were prepared from the pellets of organoids by adding plasma and thrombin in order to obtain a solid matrix. Once solidified, the organoids were fixed in 10% formalin (Thermo Scientific, 5701) and embedded in paraffin as normal tissue. Sections of 4 µm were used for IHC analyses and hematoxylin and eosin (H&E) staining (Diapath, C0303) and (Diapath, C0363), respectively. Once dried, the sections were treated with OTTIX plus solution (Diapath, X0076) and OTTIX shaper solution (Diapath, X0096) to dewax and rehydrate the sections. Antigen retrieval was performed using pH 6 solutions at 98 °C for 20 min. Next, the endogenous peroxidases and non-specific-binding sites were blocked using 3% H2O2 (VWR chemicals, 23615.248) and Protein-Block solution (DAKO Agilent technologies, X0909) respectively, for 10 min. Sections were then stained for anti-p16 (ab211542, Abcam, 1:1200), anti-Ki67 (Clone SP6; Lab Vision Corporation #RT-9106-R7, RTU) anti-Phospho-HP1γ (Ser83)

Antibody (CST #2600, 1:200), anti-CK8 (ab,59400, Abcam), anti-CK5 (ab52635, Abcam). IHC analyses were performed using the Imagescope software.

**In vivo experiments.** All animal experiments were carried out in male athymic nude mice (Balb/c nu/nu, 6–8-weeks-old), NSG mice (NOD Scid Gamma, 6–8-week-old), and NRG (NOD Rag gamma, 6–8-week-old) accordingly to protocol approved by the Swiss Veterinary Authority (No. TI-14-2014, TI-38-2018, TI-39-2018 and TI-42-2018). Patient-derived xenografts (PDX) LuCaP-147, −78, −35, −23 were provided by Eva Corey (University of Washington) and maintained as previously described[46,47]. Briefly, PDXs tumors were maintained by subcutaneous implantation of matrigel-embedded tumor fragment (1–2-mmm average diameter). In all, $2 \times 10^6$ VCaP cells, $5 \times 10^6$ LuCaP-147, LuCaP-23.1, LuCaP-35, and LuCaP-78 were resuspended in 100 µl of PBS and Matrigel 1/1 and subcutaneously injected into both of the dorsal flanks of the mice. Tumor growth was recorded using a digital caliper, and tumor volumes were calculated using the formula ($L$ x $W^2$)/2, where $L$ = length and $W$ = width of tumor. For the testosterone propionate (25 mg/kg) and SPOP inhibitor (SPOP-i, 50 mg/kg) treatment, the mice were grouped randomly, and the treatment started when the mean tumor volume reached 100 m³. Tumor volume and weight were measured two times per week. Testosterone propionate was resuspended first in ethanol (150 mg/kg) and then in Corn oil (Sigma) at a final concentration of 25 mg/kg. SPOP inhibitor was resuspended in Dulbecco's PBS at a final concentration of 50 mg/kg. At the end of the experiment, mice were euthanized, tumors extracted, and weighted. Testosterone level was measured using the Human Testosterone ELISA Kit from Abcam (ab174569). Mouse house ambient temperature was between 20 and 22 °C with humidity between 50 and 65% and a dark/light cycle of 12 h each. In order to recapitulate the levels of supraphysiological testosterone administered in clinical trials[37], mice reaching at least three times the testosterone levels measured before the treatment initiated were included in the depicted data.

**Antibodies, immunoblotting, and immunoprecipitation.** Antibodies used in immunoblotting and immunoprecipitation assays were: anti-SPOP dilution 1:1000 (ab81163, Abcam), anti-TRIM24, dilution 1:1000 (Sc-271266, Santa Cruz), anti-ß-ACTIN dilution 1:1000 (4967, Cell Signaling), anti-AR dilution 1:1000 (Sc-7305, Santa Cruz), anti-GADPH dilution 1:1000 (Sc-47724, Santa Cruz), anti-ERG dilution 1:1000 (Sc-271048, Santa Cruz), anti-α-Tubulin dilution 1:1000 (3873S, Cell Signaling), anti-ZMYND11 dilution 1:1000 (NBP2-20960, Novus Biologicals), anti-HA dilution 1:1000 (H3663, Sigma), anti-BRD2 dilution 1:1000 (A302-583A, Bethyl Labs), anti-NCOA3 dilution 1:1000 (2126, Cell Signaling), anti-DEK dilution 1:1000 (610948, BDBioscience), anti-p21 dilution 1:1000 (2947S, Cell Signaling), anti-c-MYC dilution 1:1000 (5605S, Cell Signaling), anti-HOXB13 dilution 1:1000 (Sc-28333, Santa Cruz), anti-PTEN dilution 1:1000 (9559, cell signaling), anti-p21 dilution 1:1000 (ab188224, Abcam), anti-HOXB13 dilution 1:1000 (NBP2-43655, Novus biologicals), anti-GDF-15 dilution 1:1000 (27455-1-AP, proteintech).

For immunoblotting, cells were washed with PBS and subsequently lysed in RIPA buffer (Sigma) and sonicated. Protein concentration was determined using the BCA reagent (ThermoFisher), the same amounts of protein were separated by sodium dodecyl sulfate polyacrylamide gel electrophoresis (SDS-PAGE) (Biorad) and transferred onto polyvinylidene fluoride membrane (ThermoScientific). The membrane was incubated for 1 h in 5% nonfat dry milk/TBS-T blocking buffer followed by incubation with the primary antibody overnight at 4 °C. The membrane was washed with TBS-T followed by incubation with horseradish peroxidase-conjugated secondary antibody 1:5000 (W4028 or W4018, Promega).

To detect interactions of SPOP and ZMYND11, cells were lysed in 1 % NP40 buffer (50 mM Tris-HCl pH 7.4, 150 mM NaCl, 1% NP40) with 2x protease inhibitor cocktail (Complete, Roche), sonicated, and 3 mg of lysate were incubated overnight with 2 µg of anti-HA-tag or control mouse IgG antibody (sc-2025, Santa Cruz Biotechnology) at 4 °C. Subsequently, antibodies were collected by 25 µl protein A/G magnetic beads (88803, Fisher Scientific) for 2 h, followed by two washing steps with 1% NP40 buffer. Proteins were eluted by adding 1x SDS-sample buffer under reducing conditions at 95 °C for 5 min.

**In vivo ubiquitylation assay.** In all, 293T cells were transiently transfected with indicated plasmids: pCW107-HA-ZMYND11-WT or HA-ZYMND11-DMT1/DMT2 (2 µg), pCW107-SPOP-WT or SPOP-MT (2 µg), CMV-8x Ubi-His (2 µg). Forty-two hours later, cells were treated with MG-132 (20 µM) or DMSO for additional 7 h. Cells were then washed with PBS and collected by centrifugation. A small number of cells was lysed in RIPA buffer and the rest in Buffer C (6 M guanidine–HCL, 0.1 M Na2HPO4/NaH2PO4, 10 mM Imidazole, pH = 8). The whole-cell extract was sonicated and incubated with 60 µl of Ni-NTA agarose (Sigma) overnight at 4 °C. Next, Ni-NTA beads were washed once with Buffer C, twice with Buffer D (1 volume of Buffer C: three volumes of Buffer E) and once with Buffer E (25 mMTris-HCL, 20 mM Imidazole, pH = 6.8). Elution of bound proteins was processed by boiling in 1x SDS loading buffer containing 300 mM Imidazole. Samples were loaded, separated by SDS-PAGE, and detected by immunoblotting.

**Gene expression studies**. RNA was extracted using the RNeasy kit (Qiagen) and processed by Kapa SybrFAST one-Step quantitative reverse transcription PCR (qRT-PCR) kit according to the manufacturer's instructions. qRT-PCR was undertaken on an Applied Biosystems StepOnePlus System. The target mRNA expression was quantified using ΔΔCt method and normalized to Actin expression. The primers used in the study can be found in Supplementary Table 1.

**RNA-Seq of VCaP, LNCaP, and LuCaP cells**. RNA sequencing for all experiments involving LuCaP xenografts, VCaP and LNCaP cells was performed at the Institute of Oncology Research using Next Ultra II Directional RNA Library Prep Kit for Illumina and sequenced on the Illumina NextSeq500 with single-end, 75 base pair long reads. The overall quality of sequencing reads was evaluated using various tools, namely FastQC (Andrews S., 2010), RSeQC[48], AfterQC[49] and Qualimap[50]. Sequence alignments to the reference human genome (GRCh38) was performed using STAR[51] (v.2.5.2a). Gene-expression was quantified at the gene level by using the comprehensive annotations made available by Gencode[52]. Specifically, we used *v27* release of the Gene Transfer File (GTF). Raw-counts were further processed in the *R* Statistical environment and downstream differential expression analysis was performed using the DESeq2[53] pipeline.

Genes being expressed at very low levels were automatically filtered out through the *Independent Filtering* feature embedded in DESeq2 (alpha = 0.05). Differential expression results were ranked according to the computed Wald-statistics values. Subsequently, gene-set enrichment testing was performed using Camera[54] pre-ranked (inter-gene correlation equal to 0.1, parametric test procedure). Statistical enrichments were assessed for gene-sets belonging to the Hallmark collection, which is curated by the Molecular Signature DataBase[55,56] (*MSigDB*), and for custom ERG and DHT-specific gene-signatures. All enrichments were corrected for multiple testing using Benjamini and Hochberg FDR-adjusted *P*-value.

**Identification of ERG and AR-related gene signatures**. We retrieved RNA-seq data from GEO Dataset *GSE83652*[16] to identify transcriptional perturbations in VCaP cells following treatment with DHT or following ERG silencing. To this purpose, we completely reprocessed samples SRR3713255-57, SRR3713267-72 using *STAR* and *DESeq2* as previously described for VCaP cells. In addition, to identify direct targets, we integrated information relative to AR and ERG chromatin-binding sites, which we derived from GEO Dataset *GSE28950*[27]. To maximize the number of peaks and reduce false negatives, we merged experiments performed at different time points, namely 2 h and 18 h after DHT exposure. De-multiplexed reads were aligned to *hg38* release of the human reference genome using bwa-mem[57] (0.7.15). MACS[58] (v.2.1.0) was used to perform peak calling procedure using a cutoff FDR *q*-value of 0.01 and a mappable genome size optimized for *hg38* equal to 2.9 gigabases. Downstream analysis was performed in R statistical environment. We identified binding sites overlapping promoters by using bedtools[59].

Promoters were defined as DNA regions ranging from 1500 bp upstream to 500 bp downstream of Transcription Start Sites (TSSs).

To discriminate between ERG- and AR-specific transcriptional responses we stratified genes into three main classes: genes whose promoter regions are bound by *AR* but not by *ERG*, genes whose promoters are bound by ERG but not by *AR*, and finally, genes whose promoters are co-bound by both AR and ERG. AR bound only genes were further subdivided into two sets, those being significantly (FDR < 0.05) induced following DHT treatment and those being significantly repressed. A similar approach was applied to ERG bound only genes, where genes were subdivided into ERG-induced and ERG-repressed gene-sets, if they were respectively down or upregulated following *ERG* silencing. To be more stringent in the definition of AR-specific and ERG-specific signatures, we excluded genes from the ERG-induced set that were also significantly upregulated following DHT treatment, *vice-versa* we excluded ERG-repressed genes that were significantly downregulated following DHT treatment. The same criteria were applied for DHT-specific gene-sets. Finally, defined an additional gene-set (DHT-induced/ERG-repressed) consisting of genes being co-bound by AR and ERG in their promoter region, which were significantly upregulated following DHT treatment but also significantly upregulated following ERG silencing. The overlap between custom derived gene-signatures and the most represented Hallmark's gene-sets was assessed using GeneOverlap R package[60]. Two-dimensional network visualization was generated with Cytoscape.[61]

**Gene-set testing and RNA-Seq data processing of clinical samples**. Publicly available RNA-Seq data for primary prostate cancer were obtained from The Caner Genome Atlas[3] (TCGA) database and retrieved from Genomics Data Commons (GDC) in the form of gene-centric raw counts, using *TCGAbiolinks* package[62]. We selected individuals characterized by either SPOP or ERG fusion and a third group defined as "others," which includes all remaining samples, excluding those patients exhibiting any other ETS-rearrangement. Differential expression and gene-set enrichment between samples harboring ERG fusions and SPOP-mutations were performed using *DESeq2* and *Camera* (pre-ranked) as previously described for prostate cancer cells. Single-sample gene-set enrichment analysis (GSVA[63] package) was applied to measure, for each individual patient, the overall activity of the custom gene-sets that were previously generated in VCaP cells. Following

differential expression analysis between ERG-rearranged and SPOP mutant primary tumors, we defined two gene-sets consisting of SPOP-upregulated (n = 443, log₂FC > 1, FDR < 0.05) and ERG-upregulated (n = 359, log₂FC > 1, FDR < 0.05) genes.

*PolyA* + RNASeq data for metastatic prostate cancer were obtained from SU2C cohort[64]. Normalized *RPKM* values, retrieved through *cBioportal*, were log transformed and patient's categorization (SPOP/ERG/OTHER) was performed in the same manner as for primary tumors. To evaluate whether transcriptional differences between ERG-rearranged and SPOP-mutant individuals were also conserved in the CRPC setting, we quantified the above mentioned SPOP-upregulated/ERG-upregulated signatures in the SU2C 2019 cohort, using single-sample gene-set enrichment analysis. The obtained ssGSEA scores were scaled in a range between −1 and 1 (SPOP-Upregulated) and between 1 and −1 (ERG-upregulated, inverted). Subsequently, we averaged these rescaled values in order to obtain an aggregate score.

**Circular representation of interactions between gene-sets**. Chord diagrams were generated using circlize[65] package in R statistical environment.

Strings, whose thickness is proportional to the number of shared elements, represent common genes between sets.

**ZMYND11 ChIP-seq in VCaP cells**. ChIP-seq using an anti-ZMYND11 antibody (NBP2-20960, Novus Biologicals) was performed in VCaP cells, overexpressing either wild-type SPOP or mutant SPOP harboring Y87C point mutation. Briefly, to isolate chromatin, cells (120.000.000 per IP) were cross-linked using 1% Formaldehyde cross-link protein-DNA complexes, and cross-linking was terminated by the addition of 1/10 volume 1.25 M glycine for 5 min at room temperature followed by cell lysis and sonication, resulting in an average chromatin fragment size of 200 bp. Samples lysis was performed as previously described using MNase enzyme 1000 gel units = 1 μL[66]. After adding the MNase sonication buffer, the samples were sonicated for 30 cycles, 30 s ON and 30 s OFF at high voltage. ChIP and input DNA (50 ng) were used for indexed library preparation using NEBNext Ultra II DNA Library Prep kit and subjected to 75 bp single-end sequencing on the Illumina NextSeq500. All procedures were performed at the Institute of Oncology Research. De-multiplexed reads were aligned to *hg38* release of the human reference genome using bwa-mem[57] (0.7.15). MACS[58] (v.2.1.0) was used to perform peak calling procedure using a cutoff FDR *q*-value of 0.01 and a mappable genome size optimized for *hg38* equal to 2.9 gigabases. Downstream analysis was performed in R statistical environment. ChIPseeker[67] was used to annotate peaks and to represent the distribution of ZMYND11-binding sites relative to Transcription Start Sites (TSSs). The R package chipenrich[68] was subsequently used to determine enrichment or depletion of ZMYND11 peaks in regions surrounding *TSSs* of genes that are included in Hallmarks or custom gene-set collections. Surrounding regions were defined as ranging from 5 kb upstream to 5 kb downstream of their *TSSs* (locusdef = 5 kb), which is in line with the overall behavior of ZMYND11-binding sites around *TSSs* (Supplementary Fig. 6f-g).

**Identification of AR-binding sites in primary prostate cancer specimen**. Publicly available ChIP-Seq data were retrieved from GSE120738[3]. ChIP-seq data were reprocessed as described for ZMYND11 samples. The differential binding affinity of AR between ERG-rearranged and SPOP-mutant tumors was performed using DiffBind (Stark R and Brown G, 2011).

**Frequency of SPOP mutations across patients' cohorts**. We defined the percentage of SPOP-mutant and TMPRSS2-ERG-positive tumors across different patients' cohorts originating from multiple sources. Patients with primary/loco-regional prostate tumors were derived from TCGA and MSK-IMPACT Clinical Sequencing cohorts[36]. Patients with tumor-progression (non-castrate) were derived from MSK-IMPACT and TCGA cohorts, by including from the latter only individuals that showed tumor-progression based on survival information. Castration-resistant prostate cancer patients were retrieved from MSK-IMPACT, Beltran et al.[69] and from the SU2C[64].datasets. Neuroendocrine prostate cancer samples were retrieved from the SU2C cohort (samples annotated with neuroendocrine features) and from Beltran et al.[69]. The total number of *SPOP*-mutant and *TMPRSS2-ERG* tumors was determined based on the individual studies' clinical annotations and integrated with fusion information from TCGA Fusion Gene Database (www.tumorfusions.org). Survival analysis was performed in R statistical environment using the TCGA and MSK-IMPACT clinical sequencing cohort.

**Quantitative liquid chromatography–mass spectrometry (LC-MS/MS)**. In solution digestion VCaP cell pellets were lysed at 4 °C in 8 M urea, 50 mM Tris-HCl pH 8.0, 150 mM NaCl, 1 mM EDTA, 2 μg/μl aprotinin (Sigma-Aldrich), 10 μg/μl leupeptin (Roche), and 1 mM phenylmethylsulfonyl fluoride (PMSF) (Sigma). Protein concentration was determined using a bicinchoninic acid (BCA) protein assay (Pierce). Proteins were reduced with 5 mM (DTT) for 45 min at room temperature (RT), followed by alkylation with 10 mM iodoacetamide for 30 min at room temperature in the dark. The urea concentration was reduced to 2 M using 50 mM Tris-HCl, pH 8. Samples were digested for 2 h at 25 °C with endoproteinase Lys-C (Wako Laboratories) at an enzyme-to-substrate ratio of 1:50. Samples were

subsequently digested overnight at 25 °C with sequencing grade trypsin (Promega) at an enzyme-to-substrate ratio of 1:50. Following overnight digestion, samples were acidified to a final concentration of 1% formic acid.

Peptide samples were desalted on a 100 mg tC18 Sep-Pak SPE cartridge (Waters). Cartridges were conditioned with 1 ml of 100% MeCN, 1 ml of 50% MeCN/0.1% FA, and 4x with 1 ml of 0.1% TFA. The sample was loaded, and washed 3x with 1 ml of 0.1% TFA, 1x with 1 ml of 1% FA, and eluted 2x with 600 μl of 50% MeCN/0.1% FA.

**TMT labeling of peptides**. Peptides were labeled with TMT 10-plex isobaric mass tagging reagents (Thermo Fisher Scientific). Each TMT reagent was resuspended in 41 μL of MeCN. Peptides were resuspended in 100 μL of 50 mM HEPES and combined with TMT reagent. Samples were incubated at RT for 1 h with shaking. The TMT reaction was quenched with 8 μL of 5% hydroxylamine at RT for 15 min with shaking. TMT labeled samples were combined, dried to completion, reconstituted in 100 μL of 0.1% FA, and desalted on StageTips or 100 mg SepPak columns as described above.

**Basic reverse phase (bRP) fractionation**. The TMT labeled samples were fractionated using offline high pH reversed-phase chromatography (bRP) as previously described[70]. Samples were fractionated using Zorbax 300 Extend C18 column (4.6 × 250 mm, 300 Å, 5 μm, Agilent) on an Agilent 1100 series high-pressure liquid chromatography (HPLC) system. Samples were reconstituted in 900 μL of 4.5 mM ammonium formate (pH 10) in 2% (vol/vol) acetonitrile (MeCN) (bRP solvent A). Samples were injected with Solvent A at a flow rate of 1 ml/min and separated using a 96 min gradient. The gradient consisted of an initial increase to 16% solvent B (90% MeCN, 5 mM ammonium formate, pH 10), followed by 60 min linear gradient from 16% solvent B to 40% B and successive ramps to 44% and 60% at a flow rate of 1 ml/min. Fractions were collected in a 96-deep well plate (GE Healthcare) and pooled in a non-contiguous manner into final 24 proteome fractions. Pooled fractions were dried to completeness using a SpeedVac concentrator.

**Liquid chromatography and mass spectrometry**. Desalted peptides were resuspended in 3% MeCN/0.1% FA and analyzed by online nanoflow liquid chromatography–tandem mass spectrometry (LC-MS/MS) using Q-Exactive plus mass spectrometer (Thermo Fisher Scientific) coupled online to a Proxeon Easy-nLC 1200 as previously described[70]. Briefly, 1 μg of each sample was loaded onto a microcapillary column (360 μm outer diameter × 75 μm inner diameter) containing an integrated electrospray emitter tip (10 μm), packed to ~22 cm with ReproSil-Pur C18-AQ 1.9 μm beads (Dr. Maisch GmbH) and heated to 50 °C. Samples were analyzed with 110 min LC-MS method. The 110 min method contained a mobile phase with a flow rate of 200 nl/min, comprises 3% acetonitrile/0.1% formic acid (Solvent A) and 90% acetonitrile/0.1% formic acid (Solvent B), with the following gradient profile: (min:%B) 0:2; 1:6; 85:30; 94:60; 95:90; 100:90; 101:50; 110:50 (the last two steps at 500 nl/min flow rate). The Q-Exactive plus MS was operated in the data-dependent mode acquiring HCD MS/MS scans ($r = 35,000$) after each MS1 scan ($r = 70,000$) on the 12 most abundant precursor ions using an MS1 target of $3 \times 10^6$ and an MS2 target of $5 \times 10^4$. The maximum ion time utilized for MS/MS scans was 120 ms; the HCD-normalized collision energy was set to 30; the dynamic exclusion time was set to 20 s, isotope exclusion function was enabled, and peptide match function was set to preferred. Charge exclusion was enabled for charge states that were unassigned, 1 and >6.

**MS data analysis**. All data were analyzed using Spectrum Mill software package v 6.1 pre-release (Agilent Technologies). Similar MS/MS spectra acquired on the same precursor $m/z$ within +/− 60 s were merged. MS/MS spectra were excluded from searching if they were not within the precursor MH + range of 750–4000 Da or if they failed the quality filter by not having a sequence tag length >0. MS/MS spectra were searched against UniProt human database. All spectra were allowed +/− 20 ppm mass tolerance for precursor and product ions, 30% minimum matched peak intensity, and "trypsin allow P" enzyme specificity with up to 4 missed cleavages. The fixed modifications were carbamidomethylation at cysteine, and TMT at N-termini and internal lysine residues. Variable modifications included oxidized methionine and N-terminal protein acetylation. Individual spectra were automatically designated as confidently assigned using the Spectrum Mill autovalidation module. Specifically, a target-decoy-based false-discovery rate (FDR) scoring threshold criteria via a two-step auto threshold strategy at the spectral and protein levels was used. First, peptide mode was set to allow automatic variable range precursor mass filtering with score thresholds optimized to yield a spectral level FDR of 1%. A protein polishing autovalidation was applied to further filter the peptide spectrum matches using a target protein-level FDR threshold of 0. Following autovalidation, a protein–protein comparison table was generated, which contained experimental ratios. For all experiments, non-human contaminants and reversed hits were removed. Furthermore, data were filtered to only consider proteins with two or more unique peptides and was median normalized.

**Statistical analysis**. GraphPad Prism version 8.3 (GraphPad Software) was used for statistical analysis. Data are depicted as mean + s.e.m. unless otherwise specified. The number of independent experiments or mice used is indicated in each figure legends. Unpaired Student's *t*-test was used for comparisons between two groups, one-way analysis of variance (ANOVA) with multiple comparisons for two groups or more, and two-way ANOVA with multiple comparisons for repeated measurements. Multiple comparison tests were corrected by controlling the false-discovery rate (FDR) using Benjamini and Hochberg's method. Correlation analyses were performed using Pearson correlation coefficients.

**Reporting summary**. Further information on research design is available in the Nature Research Reporting Summary linked to this article.

## Data availability

The original mass spectra have been deposited in the public proteomics repository MassIVE (identifiers MSV000082915) and are accessible at https://massive.ucsd.edu/ProteoSAFe/dataset.jsp?task=462f4fc4ab6243ac893e07ca35bd4ae3. RNA-Seq data generated have been deposited in the *ArrayExpress* database at EMBL-EBI and are accessible at: overexpression or knockdown of mutant- or wild type SPOP in VCaP prostate cancer cells https://www.ebi.ac.uk/arrayexpress/experiments/E-MTAB-7165/. Overexpression of either mutant- or wild type SPOP in presence/absence of ΔERG in LNCaP prostate cancer cells https://www.ebi.ac.uk/arrayexpress/experiments/E-MTAB-7170/. Overexpression of the SPOP-substrate ZMYND11 in VCaP cells https://www.ebi.ac.uk/arrayexpress/experiments/E-MTAB-7173/. CHiP-Seq data generated have been deposited in the *ArrayExpress* database at EMBL-EBI and are accessible at: identification of ZMYND11-binding sites in VCaP cells with stable overexpression of either mutant (Y87C) or wild-type SPOP https://www.ebi.ac.uk/arrayexpress/experiments/E-MTAB-7174/. Source data are provided with this paper.

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

## Acknowledgements

We thank the University of Washington and Eva Corey and Donna M. Peehl for providing the Patient-Derived Xenografts (PDX) models. We thank Martina Storz, Susanne Dettwiler, Carmen Gavrisan, and Christiane Mittmann for histology assistance. We thank Enrica-Mira Catò and all members of the IOR/IRB animal core facility for technical assistance and the animal work. Besides, we thank all members of the laboratory for scientific discussions. The results shown here are in whole or part based upon data generated by the TCGA Research Network: https://www.cancer.gov/tcga. J.P.T is funded by a Swiss National Science Foundation Professorship (PP00P3_150645 & PP00P3_179072) grant, and grants by the Swiss Cancer League, the Lega Ticinese contro il cancro, and the Fidinam Foundation. This work was also supported in part by grants from the National Cancer Institute (NCI) Clinical Proteomic Tumor Analysis Consortium grants NIH/NCI U24-CA210986 and NIH/NCI U01 CA214125 (to S.A.C.).

## Author contributions

T.B. and J.P.T. originally developed the concept, further elaborated on it and designed the experiments together with G.E.T. and M.B.; T.B. and G.E.T. performed experiments and analyzed the data together with N.D.U., S.A.C., A.M., T.S., L.P.B., F.S., M.Z., V.C., Anna Rinaldi, H.J., D.B., M.C., D.A., and R.G. T.B. and G.E.T. performed xenograft tumor experiments in immunodeficient mice. Z.D. and C.G.Y. provided SPOP 6b and SPOP 6c compounds. M.S., S.D., and S.M. performed immunohistochemical experiments and J.P.T the subsequent analysis. H.M. and P.S. provided prostate cancer samples. M.B., A.V., S.L., and Andrea Rinaldi analyzed genomic and RNA-Seq data. J.P.T, H.M., S.A.C., M.A.R., M.K.J., A.A., F.B., W.Z., and G.M.C provided funding and resources. J.P.T., T.B., G.E.T., and M.B. interpreted the data and wrote the paper. T.B., G.E.T., and M.B. contributed equally to this work.

## Competing interests

The authors declare the following competing interests, M.A.R. is listed as a co-inventor on US and International patents in the diagnostic and therapeutic fields of ETS gene fusion prostate cancers (Harvard and University of Michigan) and SPOP mutations (Weill Cornell Medicine). J.P.T. has received funding for the venue of scientific

conferences from Astellas, MSD, and Janssen/Cilag. The remaining authors declare no competing financial interests.
