## [Peer Review File · Nature Communications]

Reviewers' Comments:

Reviewer #2:

Remarks to the Author:

In the revised version of the manuscript "Dual functions of SPOP and ERG dictate androgen therapy responses in prostate cancer" by Bernasocchi et al. the authors still resist to show whether joint expression of both ERG and SPOP mutants in mouse prostate epithelial organoids are causal for senescence rather than cell death. At present, the conclusions are based on immunohistochemical staining data only. Do the authors also observe a decrease in Ki67 and an increase in p16, HP1gamma-phospho in WB? What about additional senescence markers such as p21, GDF15, SASP? The authors have to demonstrate that the mouse prostate organoids are derived from epithelial and not from basal cell origin. The authors statement that p16 positivity and larger cell size with cytoplasmic vacuolization may indicate the presence of a senescent cell subpopulation requests validation. The authors could easily verify their claim by performing scRNA-seq to confirm senescent cell subpopulations. Since beta-galactosidase assay, one of the gold standart experiment for measuring senescence, does not work in their organoids, it is very well possible that pathways other than senescence are leading to cell death. Such pathways are easily detected by bulk-RNAseq and scRNAseq. The authors should be able to identify a senescence signature in their mouse organoids.

Reviewer #4:

Remarks to the Author:

I believe the reviewers have largely addressed the most important points which were previously raised.

Reviewer #5:

Remarks to the Author:

Dual Functions of SPOP and ERG Dictate Androgen Therapy Responses in Prostate Cancer
Summary:

In this manuscript, Bernasocchi et al described and suggested mutant SPOP and fused ERG functioned differently in prostate tumor promotion. The co-expression of mutant SPOP and fused ERG could lead to inhibited cell proliferation and cell senescence. They further proposed the mutant SPOP pathway could stabilize ZMYND11 that in turn decrease ERG activity and increase AR signaling. In contrast, fused ERG upregulated SPOP WT results in inhibiting the AR signaling due to ZMYND11 degradation. Finally, the authors suggested the clinical implication based on two distinct mechanisms. Overall, Authors provide many interesting data and illustrate the difference the two oncogenic pathways regulated by mutant SPOP and fused ERG. However, in agreeing with critiques from two reviewers, there are some important controls are missing and some data quality is not up to NC standard. The detailed reviews are outlined below:

Major Points:

1. The manuscript writing and figure organization are not in quality shape. Lots of key data should be arranged in key figures but not in the expanding figures.
2. In extended fig 1b, why there is no endogenous ERG expression in control group?
3. Extended fig 1c, the IHC figure quality is not good enough to draw a conclusion that force

expression of mutant SPOP and fused ERG will lead to a senescence phenotype. In addition, P16 is not a core signature to determine senescence. To make an argument that mutant SPOP and fused ERG expression together lead to senescence, authors need to provide more convincing data.

4. Extended figure 1d and 1e, author mentioned that forced expression of mutant SPOP in LAPC-4 cells lead to increased proliferation with increased MYC and HOXB13 expression and decreased p21 expression. However, mutant vs control, mutant vs WT gave different results. The data presented here suggested that SPOP WT overexpression could inhibit cell growth. In addition, there is no difference between mutant vs control thus the conclusion presented here is less convincing.

5. In fig 1c and 1d, authors showed that SPOP mutant expression significantly decreased VCaP cell proliferation. However, why ERG expression level went up after in SPOP mutant expressed cells. Is this increase due to a compensatory effect? Please explain. In addition, the senescence argument is not strong here either. Why using a different set of senescence markers?

6. Extended figure 4f-4i, authors tried to establish a relationship between SPOP mutant, TRIM24 and AR signaling. They claim SPOP mutant requires TRIM24 to upregulate AR activity thus inhibit VCaP cell proliferation. However, the data presented here is not strong enough. Moreover, the western blot data in extended fig 4f does not make sense and the image quality lacks clarity. The authors suggested that they overexpressed SPOP WT and SPOP mutant in VCaP cells. However, the western blot in extended 4f showed even in the SPOP WT and SPOP mutant overexpression group, SPOP protein level varies a lot.

7. To determine the relationship between Δ ERG and SPOP and AR signaling pathway, authors used PC3 cell line which is unclear to me. Author mentioned that ERG could upregulate SPOP WT expression and thus inhibit AR signaling pathway by degrading ZMYND11. However, PC3 is an AR independent prostate cancer cells. The data here doesn't provide enough creditability to prove Δ ERG through SPOP expression to down regulate AR signaling pathway.

8. Figure 5d and extended figure 9d pattern doesn't match. High dose DHT induced cell killing in LnCap 147 in figure 5d but not in extended figure 9d.

Minor Points:

1. Overall, the manuscript figure labeling is very confusing and jumps around (e.g. Fig 1d line 141). It will be much easier for understanding if authors could rearrange the figures according to the order of manuscript.
2. Extended figure 1f dishes picture does not match the results presented in the bar graph.
3. The original blot data should contain at least two molecular weight markers.
4. Overexpression MYC lead to increased proliferation in both VCaP and LnCap-147 proves MYC's oncogenic effects but does not rule out the possibility of overexpression system is the cause. To rule out this possibility, an inducible system should be applied in here. Please revise line 90.
5. Extended Fig 2e and 2f, it is true that knock down ERG will decrease the SPOP WT expressing cells growth. However, the growth stimulation in SPOP mutant overexpressing cells are very modest. These figures only suggest that fused ERG may require SPOP WT expression. It is hard for me to believe there is an antagonistic relationship between oncogenic activation of ERG and a loss of SPOP function based on this data.
6. Line 185, what is the relationship between fig 1c with argument presented here?
7. Extended fig 4i, dishes picture does not match the results presented in the bar graph. How could a dish with no clonony (AR) with small standard deviation can be 40% of the control?

Point-by-point response to the reviewers' comments

Reviewer #2 (Remarks to the Author):

In the revised version of the manuscript "Dual functions of SPOP and ERG dictate androgen therapy responses in prostate cancer" by Bernasocchi et al. the authors still resist to show whether joint expression of both ERG and SPOP mutants in mouse prostate epithelial organoids are causal for senescence rather than cell death. At present, the conclusions are based on immunohistochemical staining data only. Do the authors also observe a decrease in Ki67 and an increase in p16, HP1gamma-phospho in WB? What about additional senescence markers such as p21, GDF15, SASP? The authors have to demonstrate that the mouse prostate organoids are derived from epithelial and not from basal cell origin. The authors statement that p16 positivity and larger cell size with cytoplasmic vacuolization may indicate the presence of a senescent cell subpopulation requests validation. The authors could easily verify their claim by performing scRNA-seq to confirm senescent cell subpopulations. Since beta-galactosidase assay, one of the gold standart experiment for measuring senescence, does not work in their organoids, it is very well possible that pathways other than senescence are leading to cell death. Such pathways are easily detected by bulk-RNAseq and scRNAseq. The authors should be able to identify a senescence signature in their mouse organoids.

Response:

We thank the reviewer for the remarks. As demonstrated by our Live/Dead viability assay (Supplementary Fig. 1d) we did not find dead cells in organoids co-expressing mutant SPOP and ERG. In addition, we included in the revised version a WB for p21 which further underscores the cell cycle arrest phenotype, previously already evidenced by p16 and ki-67. The data is also in line with corresponding findings obtained in VCaP cancer cells (Fig.1d). In the latter context, we confirm the presence of cell cycle arrest and SASP by bulk-RNAseq (Supplementary Fig. 2f and 5f).

Apart from p16 and p21, the positivity for phospho-HP1gamma – a marker for incorporation of HP1gamma into senescence-associated heterochromatin foci (SAHF) (Zhang R et al. Mol Cell Biol 2007), the vacuolization of the cytoplasm is now highlighted in greater detail (Figure 1b). Moreover, the senescence-associated beta-galactosidase (SA- β -Gal) positivity in the corresponding experiment in VCaP cancer cells (Supplementary Fig.2j) add further substance to our claims that senescence and cell cycle arrest but not cell death are the underlying cause of the synthetic sick relationship between mutant SPOP and ERG .

In the revised manuscript, we show the initial characterization of our organoids by IHC for CK5 and CK8. In accordance with the literature, the organoids display a multi-layered structure with expression of CK5 and CK8 in basal and luminal cells, respectively (Supplementary Fig.1b) (see Blattner M et al. Cancer Cell 2016, Fig. 3B).

Reviewer #4 (Remarks to the Author):

I believe the reviewers have largely addressed the most important points which were previously raised.

Response:

We thank the reviewer for the appreciation of our work.

Reviewer #5 (Remarks to the Author):

Dual Functions of SPOP and ERG Dictate Androgen Therapy Responses in Prostate Cancer Summary:

In this manuscript, Bernasocchi et al described and suggested mutant SPOP and fused ERG functioned differently in prostate tumor promotion. The co-expression of mutant SPOP and fused ERG could lead to inhibited cell proliferation and cell senescence. They further proposed the mutant SPOP pathway could stabilize ZMYND11 that in turn decrease ERG activity and increase AR signaling. In contrast, fused ERG upregulated SPOP WT results in inhibiting the AR signaling due to ZMYND11 degradation. Finally, the authors suggested the clinical implication based on two distinct mechanisms. Overall, Authors provide many interesting data and illustrate the difference the two oncogenic pathways regulated by mutant SPOP and fused ERG. However, in agreeing with critiques from two reviewers, there are some important controls are missing and some data quality is not up to NC standard. The detailed reviews are outlined below:

Major Points:

1. The manuscript writing and figure organization are not in quality shape. Lots of key data should be arranged in key figures but not in the expanding figures.

Response:

We have now rearranged the data according to *Nature Communications* standards by better highlighting our key findings into 10 main and 14 supplementary figures.

2. In extended fig 1b, why there is no endogenous ERG expression in control group?

Response:

In the prostate epithelial cells, ERG protein levels are low or undetectable. ERG protein levels are increased in prostate cancer only upon fusion between ERG and an androgen responsive gene such as TMPRSS2, as also seen by others (Park, Tomlins et al. 2010).

3. Extended fig 1c, the IHC figure quality is not good enough to draw a conclusion that force expression of mutant SPOP and fused ERG will lead to a senescence phenotype. In addition, P16 is not a core signature to determine senescence. To make an argument that mutant SPOP and fused ERG expression together lead to senescence, authors need to provide more convincing data.

Response:

We included in the revised version a WB for p21 which further underscores the cell cycle arrest phenotype, previously already evidenced by p16 and Ki-67. Apart from p16 and p21, the positivity for phospho-HP1gamma – a marker for incorporation of HP1gamma into senescence-associated heterochromatin foci (SAHF) (Zhang R et al. Mol Cell Biol 2007), the vacuolization of the cytoplasm is now highlighted in greater detail (Figure 1b). Moreover, the senescence-associated beta-galactosidase (SA-β-Gal) positivity in the corresponding experiment in VCaP cancer cells (Fig.1d, Supplementary Fig. 2j) add further substance to our claims that senescence and cell cycle arrest are the underlying cause of the synthetic sick relationship between mutant SPOP and ERG.

4. Extended figure 1d and 1e, author mentioned that forced expression of mutant SPOP in LAPC-4 cells lead to increased proliferation with increased MYC and HOXB13 expression and decreased p21 expression. However, mutant vs control, mutant vs WT gave different results. The data presented here suggested that SPOP WT overexpression could inhibit cell growth. In addition, there is no difference between mutant vs control thus the conclusion presented here is less convincing.

Response:

Indeed, SPOP-WT in the LAPC4 cells acts as a tumor suppressor by reducing growth proliferation, as expected. The differences between mutant SPOP and the control cell lines are represented by a representative image as well as a bar plot quantification highlighting at least 50% of growth advantage (statistical significance was reached for both mutants when compared to the control cell line).

5. In fig 1c and 1d, authors showed that SPOP mutant expression significantly decreased VCaP cell proliferation. However, why ERG expression level went up after in SPOP mutant expressed cells. Is this increase due to a compensatory effect? Please explain. In addition, the senescence argument is not strong here either. Why using a different set of senescence markers?

Response:

We thank the reviewer for the comment that has been previously raised by reviewer 3. Indeed, upon overexpression of mutant SPOP, ERG levels go up. This is due to both, an increase of ERG mRNA levels (as a result of increased AR signaling) and protein stability.

Below our previous answer to reviewer 3: We assessed both protein stability and mRNA expression in VCaP cells (see below). In accordance with enhanced AR signaling, ERG mRNA expression is increasing in this setting. On the other hand, we also observe an effect of mutant SPOP on ERG stability (see below). Thus, we conclude that the increase of ERG protein expression may represent the consequence of both enhanced transcription and protein stability. Despite the upregulation of ERG protein levels in the setting by mutant SPOP, its activity is not increasing but rather decreasing (current Fig. 3a). The effects are at least in part mediated by ZMYND11 upregulation (current Fig.5). See also effects of pharmacological inhibition of SPOP on ERG mRNA levels (see below).

Figure 1. a) ERG mRNA expression levels in VCaP prostate cancer cells overexpressing the indicated SPOP mutant species. b) ERG mRNA and protein expression levels in VCaP prostate cancer cells upon SPOP-i treatment. c) Immunoblots and quantification of indicated protein expression changes after treatment with cycloheximide (CHX, 100 μg/mL) in VCaP cells overexpressing the indicated SPOP species (n=2).

6. *Extended figure 4f-4i, authors tried to establish a relationship between SPOP mutant, TRIM24 and AR signaling. They claim SPOP mutant requires TRIM24 to upregulate AR activity thus inhibit VCaP cell proliferation. However, the data presented here is not strong enough. Moreover, the western blot data in extended fig 4f does not make sense and the image quality lacks clarity. The authors suggested that they overexpressed SPOP WT and SPOP mutant in VCaP cells. However, the western blot in extended 4f showed even in the SPOP WT and SPOP mutant overexpression group, SPOP protein level varies a lot.*

Response:

We have previously characterized in great detail the relationship between mutant SPOP, TRIM24, and AR signaling where we have established a function of TRIM24 as an AR coactivator (Groner A et al. Cancer Cell 2016). In line with our previous study, TRIM24 knockdown indeed reduces AR target genes in the context of SPOP-Y87C (e.g. TMPRSS2, KLK2 and PSA) and thus rescued partially the synthetic sick relationship between mutant SPOP and ERG. We provide now better western blots with less variation.

7. *To determine the relationship between ERG and SPOP and AR signaling pathway, authors used PC3 cell line which is unclear to me. Author mentioned that ERG could upregulate SPOP WT expression and thus inhibit AR signaling pathway by degrading ZMYND11. However, PC3 is an AR independent prostate cancer cells. The data here doesn't provide enough creditability to prove ERG through SPOP expression to down regulate AR signaling pathway.*

Response:

We thank the reviewer for the comment. PC3 cells have been used to assess the effects of SPOP mutations on the ERG signaling pathway independently from the AR pathway, as well as the function of SPOP WT for the oncogenic properties of ERG. We show in this system that reversal of SPOP upregulation by ERG impairs ERG-mediated tumor cell invasion. Moreover, we provide evidence that knockdown of SPOP increases AR signaling in ERG-positive VCaP cells (Supplementary Fig.10f).

8. *Figure 5d and extended figure 9d pattern doesn't match. High dose DHT induced cell killing in LnCap 147 in figure 5d but not in extended figure 9d.*

Response:

In the previous Ext Data Fig 9c (current Supplementary Fig.12c), in the clonogenic assay the DHT has been added to serum-rich medium, and the cell viability was assessed after 10 days. In contrast, in previous Fig. 5d (current Fig. 8a), the DHT dose response has been assessed in serum-free, androgen-deprived medium and for a longer period of time (two weeks). We clarified this aspect in the respective figure legends. Nevertheless, in both experimental settings there is a marked relative difference of DHT sensitivity between SPOP-mutant and ERG-mutant cancer cells.

Minor Points:

1. *Overall, the manuscript figure labeling is very confusing and jumps around (e.g. Fig 1d line 141). It will be much easier for understanding if authors could rearrange the figures according to the order of manuscript.*

Response:

In line 141, the reference to Fig.1d is a reminder of previously shown data, e.g. the immunoblot for the VCaP cells with MYC expression. We have now rearranged the data by better highlighting our key findings into 10 main and 14 supplementary figures.

2. *Extended figure 1f dishes picture does not match the results presented in the bar graph.*

Response:

The picture shown in the previous Extended Data Fig. 1f (current Supplementary Fig. 2c) are representative images of one well. The quality might have been altered upon PDF transformation and/or printing. We provide now bigger pictures.

3. *The original blot data should contain at least two molecular weight markers.*

Response:

We added a second molecular weight marker whenever possible.

4. *Overexpression MYC lead to increased proliferation in both VCaP and LnCap-147 proves MYC's oncogenic effects but does not rule out the possibility of overexpression system is the cause. To rule out this possibility, an inducible system should be applied in here. Please revise line 90.*

Response:

We thank the reviewer for the remark. We have revised line 90 (current line 92).

5. *Extended Fig 2e and 2f, it is true that knock down ERG will decrease the SPOP WT expressing cells growth. However, the growth stimulation in SPOP mutant overexpressing cells are very modest. These figures only suggest that fused ERG may require SPOP WT expression. It is hard for me to believe there is an antagonistic relationship between oncogenic activation of ERG and a loss of SPOP function based on this data.*

Response:

In the previous Extended Data Fig. 2f (current Supplementary Fig. 3f), we show a 50% viability increase when combining pharmacologic inhibition of SPOP and ERG, demonstrating a rare and statistically significant antagonistic drug effect (combining the drugs leads to a smaller effect than expected). The antagonistic relationship is further supported by orthogonal genetic experiments (current Supplementary Fig.3e).

6. *Line 185, what is the relationship between fig 1c with argument presented here?*

Response:

We thank the reviewer for this comment. In the past versions of the manuscript, Figure 1c and 1d were merged. We are here referring to the western blot in Fig. 1d and have now rectified the mistake.

7. *Extended fig 4i, dishes picture does not match the results presented in the bar graph. How could a dish with no clonony (AR) with small standard deviation can be 40% of the control?*

Response:

We thank the reviewer for this comment. We had not perform background subtraction while quantifying the cell viability through MTT assay. We apologize for this omission. We have repeated the experiment and show now better resolution pictures and corresponding quantification (Supplementary Figure 7h).

Reviewers' Comments:

Reviewer #5:

Remarks to the Author:

The revised manuscript by Bernasocchi et al has addressed good parts of my concerns and provide adequate data/explanation for most questions. However, the conclusion that force ERG and SPOP mutant expression associated senescence phenotype caused cell cycle arrest and cell death still lacks support. Data presented in this manuscript here are not strong enough to make this causal relationship argument. In addition, the revised P21 data looks very similar to the original one. The different senescence markers used in different figure also raised my concern for their senescence conclusion. Thus, authors need to revise their manuscript to summarize that senescence could be one possible reason leading to the observed phenotype. If authors can edit the manuscript accordingly, I would recommend this manuscript for publication in Nature Communication.

Point-by-point response to the reviewers' comments

Reviewer #5 (Remarks to the Author):

The revised manuscript by Bernasocchi et al has addressed good parts of my concerns and provide adequate data/explanation for most questions. However, the conclusion that force ERG and SPOP mutant expression associated senescence phenotype caused cell cycle arrest and cell death still lacks support. Data presented in this manuscript here are not strong enough to make this causal relationship argument. In addition, the revised P21 data looks very similar to the original one. The different senescence markers used in different figure also raised my concern for their senescence conclusion. Thus, authors need to revise their manuscript to summarize that senescence could be one possible reason leading to the observed phenotype. If authors can edit the manuscript accordingly, I would recommend this manuscript for publication in Nature Communication.

Response:

We appreciate and agree with the reviewer's comments. We modified the text (line 83-90;) according to his suggestion, and now it states: “[...] Similarly, to the mouse prostate organoids, **possible** induction of senescence was evidenced by increased senescence-associated β -galactosidase (SA- β -gal) positive cells and upregulation of p21 and GDF15 protein levels. In line with this, the transfer of conditioned medium from VCaP cells expressing mutant SPOP (SPOP-Y87C, SPOP-W131G) also reduced the proliferation of parental VCaP cells, indicating a **possible** contribution of Senescence-Associated Secretory Phenotype (SASP) **and suggesting that senescence could be one of the possible biological pathways involved in the synthetic sick relationship.**”